# Wag31, a membrane tether, is crucial for lipid homeostasis in mycobacteria

Yogita Kapoor[1,2], Himani Khurana[3†], Debatri Dutta[1†], Arnab Chakraborty[3], Anshu Priya[2,4], Archana Singh[2,4], Siddhesh S Kamat[3], Neeraj Dhar[5,6], Thomas John Pucadyil[3], Vinay Kumar Nandicoori[1,2,7]*

[1]CSIR-Centre for Cellular and Molecular Biology (CSIR-CCMB), Hyderabad, India; [2]Academy of Scientific and Innovative Research (AcSIR), Ghaziabad, India; [3]Indian Institute of Science Education and Research Pune, Pune, India; [4]CSIR-Institute of Genomics and Integrative Biology (CSIR-IGIB), New Delhi, India; [5]Vaccine and Infectious Disease Organisation, University of Saskatchewan, Saskatoon, Canada; [6]Department of Biochemistry, Microbiology and Immunology, University of Saskatchewan, Saskatoon, Canada; [7]National Institute of Immunology, Delhi, India

## eLife Assessment

Understanding bacterial growth mechanisms potentially uncover novel drug targets which are crucial for maintaining cellular viability, particularly for bacterial pathogens. In this **important** study, Kapoor et al, investigate the role of Wag31 in lipid and peptidoglycan biosynthesis in mycobacteria. A detailed analysis of Wag31 domain architecture revealed a role in membrane tethering. More specifically, the N-terminal and C-terminal domains appeared to have distinct functional roles. The data presented are **solid** and support the conclusion made. This study will be of broad interest to microbiologists and molecular biologists.

*For correspondence:
vinaykn@nii.ac.in

†These authors contributed equally to this work

**Abstract** The mycobacterial cytoskeletal protein Wag31 is necessary for maintaining cell shape and directing cellular growth and elongation. Wag31 has a characteristic N-terminal DivIVA-domain and a C-terminal coiled-coil domain. While the role of Wag31 in polar elongation is known, there is limited mechanistic insight on how it orchestrates growth and elongation. In this report, we delineate roles of the N- and C-terminal domains of Wag31 using genetics, state-of-the-art multi-omics, biochemical, and imaging approaches. We show that Wag31 predominantly interacts with several membrane-associated proteins involved in lipid metabolism, cell wall synthesis, and division. Native levels of Wag31 are critical for the maintenance and distribution of membrane lipids. Both depletion and overexpression of Wag31 perturb lipid homeostasis, leading to the formation of intracellular lipid inclusions (ILIs). Protein–lipid crosslinking and imaging studies reveal that purified Wag31 can bind and effectively tether cardiolipin (CL)-containing liposomes. We further show that the tethering activity lies in the DivIVA-domain containing N-terminal of Wag31 while the C-terminal mediates protein–protein interactions of Wag31. Despite retaining its ability to interact with partner proteins, the DivIVA-domain-deleted Wag31 mutant shows defects in liposome tethering in vitro and non-polar localization of CL in vivo, which eventually causes lethality. Our study suggests that membrane tethering 'licenses' Wag31 to form scaffolds that help orchestrate protein–lipid and protein–protein interactions necessary for mycobacterial growth and survival.

## Introduction

Cell division is a critical event in the life of a bacterium as it is the prime mode of bacterial growth and reproduction. Hence, the process of their division is tightly knitted. The key process in bacterial cell division, that is, assembly of the FtsZ ring at the mid-cell (*Bi and Lutkenhaus, 1991*), is tightly regulated by the Min and Nucleoid occlusion systems (*de Boer et al., 1991*; *Hu et al., 1999*; *Raskin and de Boer, 1997*; *Meinhardt and de Boer, 2001*; *Woldringh et al., 1990*). Mycobacterium, which comprises clinically relevant pathogens – *Mycobacterium tuberculosis* (*Mtb*), *Mycobacterium leprae*, and *Mycobacterium abscessus* lacks both the regulators of FtsZ ring placement (*Logsdon and Aldridge, 2018*). Instead, they encode for a DivIVA protein, usually encoded by Firmicutes and other Actinobacteria (*Cha and Stewart, 1997*; *Edwards and Errington, 1997*; *Kang et al., 2008*; *Jani et al., 2010*). It is necessary that the bacterial shape, crucial for normal cellular physiology, is maintained during cell division. Rod-shaped bacteria such as *Escherichia coli* and *Bacillus subtilis* encode MreB and multiple MreB-like proteins that sustain the rod morphology of cells (*Letek et al., 2008*). *Mycobacterial* spp. lack MreB homologues (*Thanky et al., 2007*; *Kieser and Rubin, 2014*). The DivIVA protein of *Mtb* called Wag31 is known to regulate both cellular growth and morphology (*Kang et al., 2008*; *Nguyen et al., 2007*), which makes it an exciting candidate to investigate.

DivIVA proteins were discovered three decades ago during the study of cell division mutants of *B. subtilis* (*Hammond et al., 2019*). DivIVA proteins are rich in coiled-coil domains (*Stahlberg et al., 2004*; *Muchová et al., 2002*; *Letek et al., 2009*; *Rigden et al., 2008*; *Choukate et al., 2020*), which enable them to form higher oligomeric structures and, hence, scaffolds, making them suitable adaptor proteins (*Flärdh, 2003*; *Fadda et al., 2007*). They are known to play diverse roles in various species (*Kaval and Halbedel, 2012*), for example, they play dual roles of septum positioning and chromosome partitioning in *B. subtilis* (*Edwards and Errington, 1997*; *Thomaides et al., 2001*); cell separation/resolution post-cell-division in *Listeria monocytogenes* (*Kaval et al., 2017*; *Kaval et al., 2014*); pole formation and maturation in *Streptococcus* spp. (*Fadda et al., 2007*). In *Mycobacterium* spp., Wag31 is an essential protein that senses negative membrane curvature found at poles and septum and localizes there (*Meniche et al., 2014*). It plays roles in cellular elongation and septation in concert with other divisome and elongasome components (*Kang et al., 2008*; *Nguyen et al., 2007*; *Melzer et al., 2018*). It is specifically found in higher amounts at the old pole, where it directs polar elongation by recruiting the acyl CoA carboxylase (ACCase) complex involved in fatty acid, mycolic acid, and methyl-branched chain lipid precursor synthesis (*Meniche et al., 2014*; *Xu et al., 2014*; *Hannebelle et al., 2020*; *Santi et al., 2013*). It localizes to the septum after cytokinesis is over, perhaps for septal synthesis via its interactions with FtsI (*Santi et al., 2013*; *Mukherjee et al., 2009*; *Habibi Arejan et al., 2022*), though the process remains largely unknown. Consequently, loss of Wag31 leads to cell division defects – loss of polar elongation, which renders the cells 'round' in shape that occurs due to affected peptidoglycan (PG) synthesis (*Kang et al., 2008*; *Nguyen et al., 2007*; *Meniche et al., 2014*). Though Wag31 participates in coordinating PG metabolism, the mechanism or the protein network by which it does so remains unknown.

Wag31 has also been associated with the maintenance of the intracellular membrane domain (IMD), a specialized compartment for lipid synthesis, localized predominantly at the old pole in consonance with pole elongation and spottily all around the plasma membrane where it may function to repair membrane defects (*Morita et al., 2005*; *Hayashi et al., 2016*; *García-Heredia et al., 2021*). The loss of Wag31 leads to the delocalization of IMD and its protein markers, such as MurG and GlfT2, but it is not understood whether Wag31 plays a direct role in membrane partitioning or it is just a consequence of localized membrane synthesis (*Habibi Arejan et al., 2022*; *García-Heredia et al., 2021*). The role of Wag31 in PG and lipid synthesis that drives cell division remains unclear. Despite numerous studies, the details of how Wag31 orchestrates mega-complexes at the pole or partitions the membrane into two to govern cellular elongation remain elusive.

Here, we endeavoured to address the lacunae in the functionality of Wag31 by asking the following questions: (1) what are the ultrastructural changes that occur in mycobacteria due to the absence of Wag31, (2) do Wag31-deficient cells undergo transcriptional rewiring to adjust to cell division defects, (3) is the change in cellular morphology from 'rod-to-round' accompanied by altered lipid levels, (4) what is the interactome of Wag31, (5) what are the functions of its N- and C-terminal domains, and finally (6) what is its relationship with the mycobacterial membrane. The present study shows that Wag31 levels are critical for maintaining lipid homeostasis in mycobacteria. Our results demonstrate

that the N- and C-terminals of Wag31 have distinct functions. The N-terminal of Wag31 binds and tethers the membrane together while the C-terminal interacts with various proteins. Together, the domains help Wag31 to orchestrate membrane-localized processes essential for maintaining normal cellular morphology and physiology.

## Results

### Loss of Wag31 leads to the formation of ILIs

In the pursuit of bridging the knowledge gap existing in the functionality of Wag31 (*Figure 1a*), we generated a *Wag31* conditional mutant in *Msm*, by integrating a copy of *wag31* under tetracycline (ATc- tet off) regulation at the L5 locus and replacing the native copy with a hygromycin cassette (*Figure 1—figure supplement 1a*). The mutant, *Δwag31*, thus generated, was confirmed by PCRs (*Figure 1—figure supplement 1b*) and western blotting (*Figure 1b*). The lack of Wag31 led to a drop in the turbidity of culture (*Figure 1c*). Analysis of bacillary survival in vitro by enumerating CFUs 12 hr post-ATc addition, revealed a 2 log fold difference in the survival of *Δwag31* (*Figure 1d*). Given the essential role of Wag31 in causing polar growth of the cell, we investigated the effect of depletion of Wag31 on bacterial cellular morphology using electron microscopy (EM). Scanning EM (SEM) suggested that the absence of Wag31 caused a transition in the morphology of the cells from rod to round, as previously documented (*Kang et al., 2008*; *Nguyen et al., 2007*; *Singh et al., 2017*). Apart from round morphology (~36 %), we also observed a rod-to-round transformation stage in which cells were bulged at one pole with the other pole intact (*Figure 1e–g*). To obtain detailed insights, we performed transmission EM (TEM), which divulges information on the internal structure of cells. We discovered spherical, electron-lucent bodies (indicated by an arrowhead in *Figure 1h*) in the cytosol of all the samples. While *Msm* and *Δwag31*-ATc harboured a few (~2–3) electron-lucent bodies, their numbers were drastically higher in the case of *Δwag31*+ATc (*Figure 1h*). The cytosol of *Δwag31*+ATc was filled with multiple variable-sized bodies which resembled lipid bodies that *Mtb* accumulates inside the host (*Daniel et al., 2011*; *Barisch et al., 2015*). As the term 'lipid bodies/lipid droplets' is used specifically for the lipids acquired by *Mtb* from the host cell, we referred to them as ILIs. We categorized the cells based on the number of ILIs they contained. Their distribution remained reasonably similar within *Msm* and *Δwag31*-ATc cells for all the classes (*Figure 1i*). Whereas *Δwag31* treated with ATc had as high as 85% cells harbouring more than five ILIs and very few cells with less than five ILIs (*Figure 1i*). Our data, therefore, report a novel observation of lipid accumulation when Wag31 is depleted from mycobacteria.

### Wag31 levels influence lipid homeostasis

Next, we set out to analyse if the ILIs are indeed lipidic in nature by staining cells with BODIPY (493/503), a dye that preferentially partitions into neutral lipid droplets (*Bartz et al., 2007*; *Tauchi-Sato et al., 2002*). We stained either untreated or ATc-treated *Msm* and *Δwag31* with BODIPY and visualized them on a microscope. ATc-treated *Δwag31* cells displayed the highest level of BODIPY fluorescence, indicating the maximum abundance of lipid bodies among the three strains (*Figure 2a, c*). Further, we examined whether the appearance of ILIs is associated just with the loss of Wag31 or is a consequence of deviations from the native expression level of Wag31. Towards this, we overexpressed Wag31 episomally using an isovaleronitrile (IVN) inducible system (*Msm::wag31*) (*Figure 2—figure supplement 1a*) and stained the cells with BODIPY. Unlike Wag31 depletion, which resulted in spherical cells (*Figure 1e*), Wag31 overexpression resulted in the bulging of cells from one pole (*Figure 2b*). Additionally, the overexpression of Wag31 also led to lipid aggregates in the cell with a few larger 'donut-like' aggregates in the bulged pole (*Figure 2b, d*).

Lipid accumulation in both cases led us to inspect whether lipid metabolism is closely tied to Wag31 expression levels in mycobacteria. To investigate that possibility, total cellular lipids were extracted from *Msm*, Wag31 depletion, and overexpression conditions and subjected to semi-quantitative LC–MS analysis (illustrated in *Figure 2e*). Lipidomic analysis of *Δwag31*+ATc revealed an imbalance in the levels of neutral lipids, phospholipids, and free fatty acids in comparison to *Msm*-ATc (*Figure 2f, g*). *Δwag31*+ATc cells had a significant accumulation of unsaturated diacylglycerols (~3- to 5-fold) and a very long chain containing (>C36) phosphatidylethanolamines (PE; ~5- to 32-fold). This was accompanied by increased levels of different chain lengths (C18–C24) of free fatty acids (~1.7- to 4-fold) and

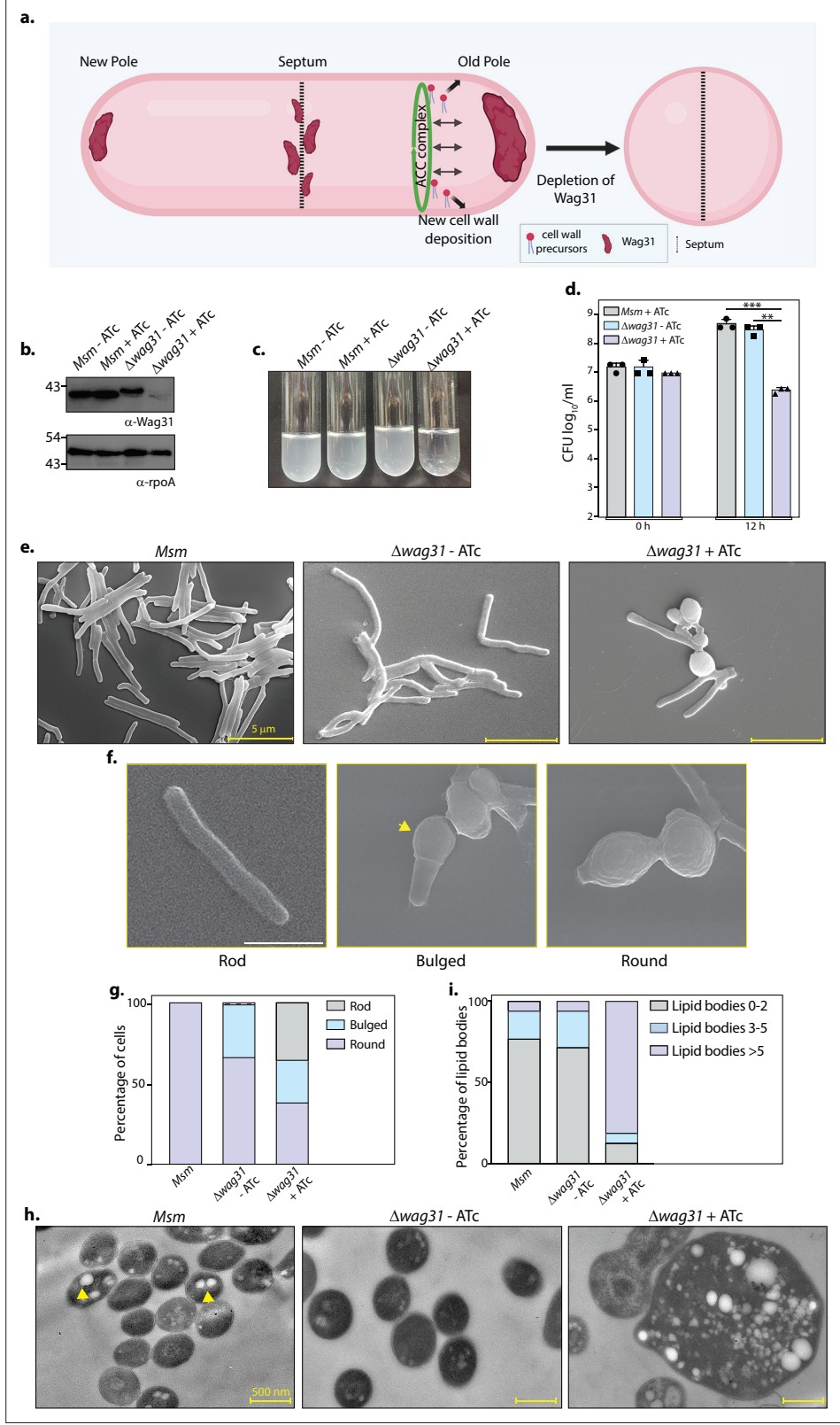

**Figure 1.** Loss of Wag31 leads to the formation of intracellular lipid inclusions (ILIs). (**a**) Illustration depicting the known role of Wag31 in directing polar elongation and maintaining rod-shaped morphology of cells. Created in BioRender.com. (**b**) Whole-cell lysates (WCL) were prepared from *Msm* and *Δwag31* either untreated or treated with 100 ng/ml ATc for 12 hr, and 40 μg lysate was resolved on a 12% gel, transferred on nitrocellulose membrane,

*Figure 1 continued on next page*

*Figure 1 continued*

and probed with either α-Wag31 or α-rpoA antibody (loading control), respectively. (**c**) Cultures showing growth of *Msm* and *Δwag31* either untreated or treated with 100 ng/ml ATc for 12 hr. (**d**) *Msm*+ATc, *Δwag31*-ATc, and *Δwag31*+ATc were tracked for bacillary survival at 0 and 12 hr by enumerating CFUs. Appropriate serial dilutions were plated on 7H11 plates (without antibiotic or ATc), and bar graphs demonstrating bacillary survival (CFU $\log_{10}$/ml ± standard deviations (SD), represented by error bars) were plotted at indicated time points. Statistical significance was calculated using two-way ANOVA 0.0021 (**), 0.0002 (***). The experiment was performed twice independently, each performed in triplicates. (**e**) Scanning electron micrographs (SEM) showing *Msm*, *Δwag31*-ATc, and *Δwag31*+ATc at 12 hr post-treatment. (**f**) Cells depicting rod, bulged, or round morphology. The yellow arrowhead indicates bulged pole of the cell. (**g**) 200 cells from each group were checked for either rod, bulged, or round morphology, and the percentage of cells with these morphologies in all the three strains was plotted in a bar graph. (**h**) Transmission electron micrographs (TEM) showing *Msm*, *Δwag31*-ATc, and *Δwag31*+ATc at 12 hr post-treatment. (**i**) 250 cells, each from *Msm*, *Δwag31*-ATc, and 185 cells from *Δwag31*+ATc, were divided into three classes (0–2, 2–5, and >5 ILIs) based on the number of ILIs and the percentage of each class in every strain was plotted in a bar graph.

The online version of this article includes the following source data and figure supplement(s) for figure 1:

**Source data 1.** Source data used for *Figure 1b, c, e, f, h*.

**Source data 2.** Unmarked and uncropped source data used for *Figure 1b, c, e, f, h*.

**Source data 3.** Source data used for plotting *Figure 1d, g, i*.

**Figure supplement 1.** Generation of Wag31 knockdown.

**Figure supplement 1—source data 1.** Source data used for *Figure 1—figure supplement 1b*.

**Figure supplement 1—source data 2.** Unmarked and uncropped source data used for *Figure 1—figure supplement 1b*.

reduced levels of cardiolipins C70:0 and C70:1 (~3-fold). Wag31 overexpression, on the other hand, led to a twofold increase in the level of different species of alpha-mycolic acids (α-MA; C74–C78) as compared to Msm+IVN (*Figure 2h*), while other tested lipids remained unchanged (*Figure 2—figure supplement 1b–g*). Altogether, the results presented above demonstrate that Wag31 levels are crucial for maintaining lipid homeostasis in mycobacteria, the depletion and overexpression of which impact lipid metabolism differently.

## Molecular interactions of Wag31 with membrane-associated proteins govern cellular homeostasis

We reasoned that Wag31, an adaptor protein, might maintain lipid homeostasis by interacting with proteins involved in these pathways. To examine the protein interaction network of Wag31, we performed an interactome analysis by utilizing *Δwag31*, which expresses FLAG-Wag31 integrated at the L5 locus, and used FLAG-GFP as the control to differentiate from non-specific interactions. FLAG-Wag31 and FLAG-GFP from biological quadruplets were immunoprecipitated using whole-cell lysates (WCLs) from the log phase cells and subjected to mass spectrometry analysis (*Figure 3—figure supplement 1a*). This study identified both direct and indirect protein partners of Wag31. The proteins identified were categorized based on their functions. It was found that Wag31 interacts with proteins involved in cell-wall elongation and division, lipid metabolism, intermediary metabolism, and respiration and information pathways (translation and replication) (*Figure 3a*, *Figure 3—figure supplement 2*). Interactions of Wag31 with AccA3 and AccD5, subunits of the ACCase complex, and Rne or RNaseE (Msm4626), a membrane-associated protein, have been reported previously (*Meniche et al., 2014*; *Griego et al., 2022*). Subunits of ACCase complex and Rne were among the top 5 hits in our MS/MS data, validating our approach (*Figure 3a*, marked with asterisk). The classification of the top 20 hits is represented in *Figure 3a*. The highest number of protein interactors, expectedly, fell in the cell wall and cell processes category (*Figure 3a*). For the present study, we focussed on the interactors found in the cell wall elongation and cell division and lipid metabolism functional categories to probe the cause behind the lipid dysbiosis phenotype.

MurG, involved in PG metabolism (*Hayashi et al., 2016*; *Priyadarshini et al., 2007*; *Heidrich et al., 2001*; *Mengin-Lecreulx et al., 1991*), and SepIVA, involved in regulating septation and the spatial regulator of MurG (*Freeman et al., 2023*) also surfaced in the interactome of Wag31. Msm1813, or acyl CoA carboxylase (ACAD), crucial for nascent fatty acid precursor synthesis, was also

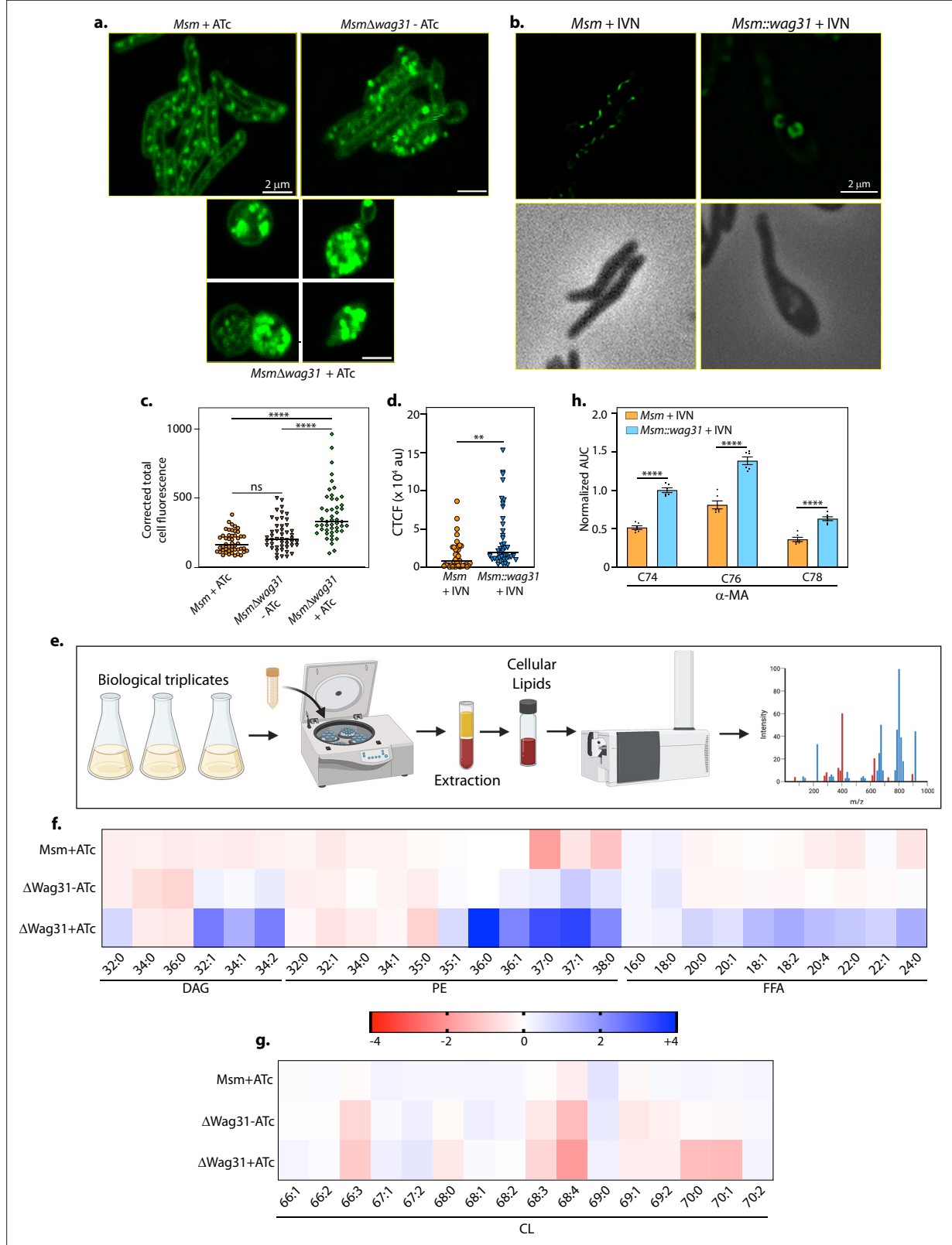

**Figure 2.** Wag31 levels perturb lipid homeostasis. Representative micrographs showing BODIPY staining in (**a**) *Msm*+ATc, *Δwag31*-ATc, and *Δwag31*+ATc samples imaged 12 hr post-ATc treatment. (**b**) Msm+IVN, *Msm::wag31*+IVN samples imaged 12 h post 5 µM IVN treatment. (**c**) Corrected total cell fluorescence was calculated in individual bacterial cells from Msm+ATc, *Δwag31*-ATc, and *Δwag31*+ATc strains (*N* = 48) and plotted in a graph. Horizontal bar represents median corrected total cell fluorescence (CTCF). Statistical analysis was performed using one-way ANOVA followed by Brown–

*Figure 2 continued on next page*

*Figure 2 continued*

Forsythe test (Tukey's multiple comparison test; 0.0021 (**),<0.0001 (****)). (**d**) Corrected total cell fluorescence was calculated for 42 individual cells from Msm+IVN, and *Msm::wag31*+IVN strains and plotted in a graph (statistical analysis was performed using Welch's *t*-test 0.0021 (**)). (**e**) Schematic depicting the workflow of Lipidomics (LC–MS). Created in BioRender.com. (**f, g**) Heatmap depicting log$_2$ fold change (FC) in indicated lipid classes of Msm+ATc, *Δwag31*-ATc, and *Δwag31*+ATc as compared to *Msm*-ATc. The experiment was performed with six biological replicates, and the abundances of identified lipid classes were normalized to *Msm*-ATc to yield relative abundances of lipids in other strains. Blue and red colours indicate up- and downregulated lipid classes; colour intensity variation changes with log$_2$ FC. (**h**) Bar graph (mean ± SEM) depicting the abundance of α-MA in *Msm* and *Msm::wag31* treated with 5 µM IVN for 12 hr. Values on the *y*-axis represent area under the curve (AUC) normalized to an internal standard (FFA 15:0). *x*-axis represents various species (chain length variations) of α-MA (statistical analysis was performed using two-way ANOVA followed by a Tukey's multiple comparison's test; <0.0001 (****)).

The online version of this article includes the following source data and figure supplement(s) for figure 2:

Source data 1. Source data used for *Figure 2a, b*.

Source data 2. Unmarked and uncropped source data used for *Figure 2a, b*.

Source data 3. Source data used for plotting *Figure 2c, d, f–h*.

Figure supplement 1. Wag31 levels perturb lipid homeostasis.

Figure supplement 1—source data 1. Source data used for making *Figure 2—figure supplement 1a*.

Figure supplement 1—source data 2. Unmarked and uncropped source data used for making *Figure 2—figure supplement 1a*.

Figure supplement 1—source data 3. Source data used for plotting *Figure 2—figure supplement 1b–d, f, g*.

enriched in the interactome. Other lipid metabolism interactors of Wag31 included acyl-CoA dehydrogenase (Msm2080), involved in lipid catabolism; GpsA, involved in phospholipid biosynthesis; and Msm0372, involved in fatty acid biosynthesis. To validate the high-throughput data, we chose MurG, SepIVA, and Msm2092 and performed pulldown assays with His-tagged Wag31. His-tagged Wag31 (*Figure 3b*) was incubated with *Msm* lysates expressing 3X-FLAG tagged versions of the interactors (*Figure 3c*) and pulled-down with the help of cobalt beads followed by a western blot analysis. Results obtained confirmed the interaction of Wag31 with MurG, SepIVA and Msm2092 (*Figure 3d*). MmpL4 and MmpS5 which did not appear in the interactome database were used as negative controls in this experiment and a similar sassy was performed with them (*Figure 3—figure supplement 1b–d*). Western blot analysis did not show an interaction of Wag31 with either MmpL4 or MmpS5 (*Figure 3—figure supplement 1d*) thus validating the interactome data.

Since Wag31 is a spatiotemporal regulator of other proteins, we were intrigued to discover the localization of its interacting partners. We used the data curated by *Zhu et al., 2021* and determined that Wag31 majorly interacts with membrane-associated proteins with a few exceptions (*Figure 3e*, *Figure 3—figure supplement 1e*). The mycobacterial plasma membrane has been recently shown to have a cell wall-associated compartment called PM-CW and a cell wall-free compartment called PM$_f$ or IMD, which is the site for lipid synthesis. Based on the available data (*Hayashi et al., 2016*; *Zhu et al., 2021*; *He and De Buck, 2010*), we found that Wag31 interacts more with IMD proteins than with PM-CW proteins (*Figure 3f.*) In accordance with assigned roles (*Hayashi et al., 2016*), PM-CW residing proteins in the interactome are involved in cell wall and cell processes and intermediary metabolism and respiration. Similarly, IMD-homing proteins clustered in the lipid metabolism and cell wall synthesis category. These results suggest that Wag31 maintains lipid homeostasis and cellular morphology primarily through its interactions with the membrane proteins participating in such pathways.

## Wag31 binds and tethers CL-containing membranes

Wag31 is a cytoplasmic protein that recognizes negative curvature regions in the cell, that is, poles and septum (*Meniche et al., 2014*). It also forms higher-order structural assemblies at the poles (*Choukate et al., 2020*; *Choukate and Chaudhuri, 2020*). However, the precise mechanism by which Wag31 assemblies localize to the poles remains elusive. The predominant interactions of Wag31 with membrane-associated proteins (*Figure 3*) were intriguing and led us to investigate if it can directly bind to membranes. We reasoned that such membrane-binding activity might render it capable of acting as a scaffold that anchors and organizes other proteins, thereby modulating their functions. To test this, we performed a lipid crosslinking-based proximity-based labelling of membrane-associated protein (PLiMAP) assay (*Figure 4a*). Purified Wag31 was incubated with ~100 nm extruded liposomes

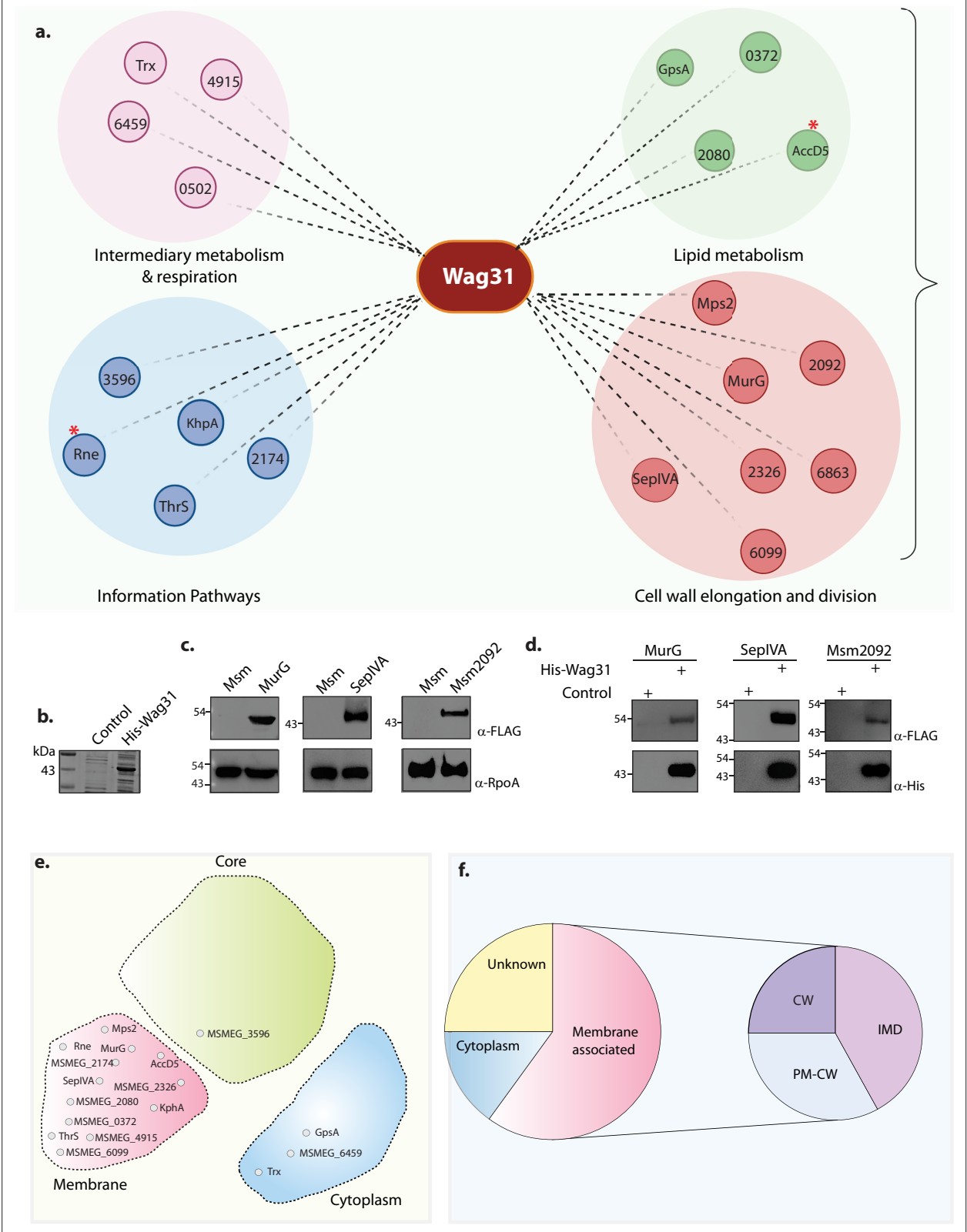

**Figure 3.** Molecular interactions of Wag31 with membrane-associated proteins govern cellular homeostasis. (**a**) Illustration showing the distribution of the top 20 interactors of Wag31 identified from MS/MS analysis from three independent immunoprecipitations. Interactors were classified on the basis of their functional category based on data available in Mycobrowser (**Kapopoulou et al., 2011**). MSMEG numbers or the identity of interactors are marked in smaller circles. Created in BioRender.com. (**b**) Coomassie Brilliant blue (CBB) staining of plain *E. coli* lysate and *E. coli* lysate expressing

*Figure 3 continued on next page*

*Figure 3 continued*

His-Wag31. *E. coli* lysate prepared from BL21-DE3 strain was used as the control. (**c**) Representative western blots demonstrating expression of 3XFLAG tagged-MurG, -SepIVA, and -Msm2092. Whole-cell lysates (WCLs) were prepared from 0.5 µM IVN induced *Msm::murG*, *Msm::sepIVA*, *Msm::msm2092* and 40 µg of each was resolved on a 10% SDS–PAGE, transferred onto a nitrocellulose membrane and probed with α-FLAG and α-RpoA (loading control). (**d**) Representative western blots showing interaction of MurG, SepIVA, and, Msm2092 with Wag31. *Msm* and *E. coli* lysates represented in (**b, c**) incubated together and pulled down with Cobalt beads were resolved on a 10–12% SDS–PAGE, transferred onto nitrocellulose membrane and probed with α-His to detect the pulldown and α-FLAG to detect the interaction. The experiment was performed independently twice. (**e**) Graphic representation (adapted from Figure 3E from *Zhu et al., 2021*) shows the distribution of the top 20 interactors in different compartments of a mycobacterial cell: membrane, cytoplasm, or core (involved in DNA replication and recombination). Interactors from each compartment are marked inside the respective domain. (**f**) Pie chart shows the distribution of interactors belonging to the IMD, PM-CW, or membrane proteins whose specific localization is not determined. Percentage of proteins (top 20 hits) belonging to each membrane compartment is mentioned.

The online version of this article includes the following source data and figure supplement(s) for figure 3:

**Source data 1.** Source data used for generating *Figure 3a, e, f*.

**Source data 2.** Unmarked and uncropped source data used for generating *Figure 3a, e, f*.

**Source data 3.** Source data used for generating *Figure 3b–d*.

**Figure supplement 1.** Molecular interactions of wag31 with membrane proteins govern cellular homeostasis.

**Figure supplement 1—source data 1.** Source data used for generating *Figure 3—figure supplement 1a–d*.

**Figure supplement 1—source data 2.** Unmarked and uncropped source data used for generating *Figure 3—figure supplement 1a–d*.

**Figure supplement 2.** Molecular interactions wag31 with membrane proteins govern cellular homeostasis.

**Figure supplement 2—source data 1.** Source data used for generating *Figure 3—figure supplement 2*.

containing each of the abundant classes of bacterial lipids PE, phosphatidylglycerol (PG), or CL. Liposomes contained trace amounts of a reactive fluorescent lipid, which crosslinks with membrane-bound proteins upon photoactivation. These assays revealed that Wag31 preferentially binds CL- but not PG- or PE-containing liposomes (*Figure 4b*). Both CL and PG are anionic lipids, and the preferential binding of Wag31 to CL indicates that the binding cannot be attributed to a charge-based interaction. Remarkably, the sensitive fluorescence-based detection revealed the presence of SDS-resistant higher-order oligomers of membrane-bound Wag31 (*Figure 4b*).

Wag31 is a homologue of the DivIVA protein from *B. subtilis*. The latter has been shown to bind to the membrane via the N-terminal DivIVA-domain (N-DivIVA$_{BS}$) (*Oliva et al., 2010*). However, the functional significance of the N- and C-terminal domains of mycobacterial Wag31 has not been dissected. Considering that the Wag31 N-terminal region shares 42% sequence identity with N-DivIVA$_{BS}$ (*Choukate and Chaudhuri, 2020*), we sought to inspect the effect of disruption of a particular residue (K20) in the N-terminus and the deletion of the entire N-terminus, encompassing the DivIVA-domain (1–60 amino acids) on membrane binding (*Figure 5e*). The K20 residue lies within a positively charged patch, which likely constitutes a membrane-associating surface in the intertwined loop region of Wag31 (*Choukate and Chaudhuri, 2020*). Moreover, mutating the K20 residue has been shown to decrease the cell length and elongation rate by 18% and 22%, respectively (*Habibi Arejan et al., 2022*). The K20A mutation showed a marginal defect in binding to CL liposomes (*Figure 4c, d*), indicating that K20 alone is insufficient for membrane binding. However, Wag31$_{K20A}$ showed a marked defect in forming SDS-resistant higher-order oligomers, as seen in the binding assays. Interestingly, deletion of the entire N-terminal domain caused a significant defect in both membrane binding and forming SDS-resistant higher-order oligomers (*Figure 4c, d*). Surprisingly, binding was not entirely abolished, suggesting that the C-terminus of Wag31 may also contribute to membrane binding. Taken together, our results indicate that Wag31 preferentially binds CL-containing membranes and that regions other than the N-terminal domain contribute to membrane binding. The latter feature distinguishes Wag31 from DivIVA$_{BS}$, and Wag31 likely contains membrane binding sites located at both the N- and C-terminal regions, which might lead to differences in the manner in which it interacts with the membrane and partner proteins compared to DivIVA$_{BS}$.

To further confirm membrane binding, we imaged 100 nm extruded CL-containing liposomes with a membrane dye DiI in the presence of Wag31. We used a GFP-tagged Wag31 mixed with an untagged Wag31 at a 1:10 molar ratio for these experiments. In the absence of Wag31-GFP, liposomes appeared as faint but distinct fluorescent puncta (*Figure 4e*, first panel). Surprisingly, in reactions containing Wag31-GFP, we observed liposomes clustered into bright fluorescent puncta that were positive for

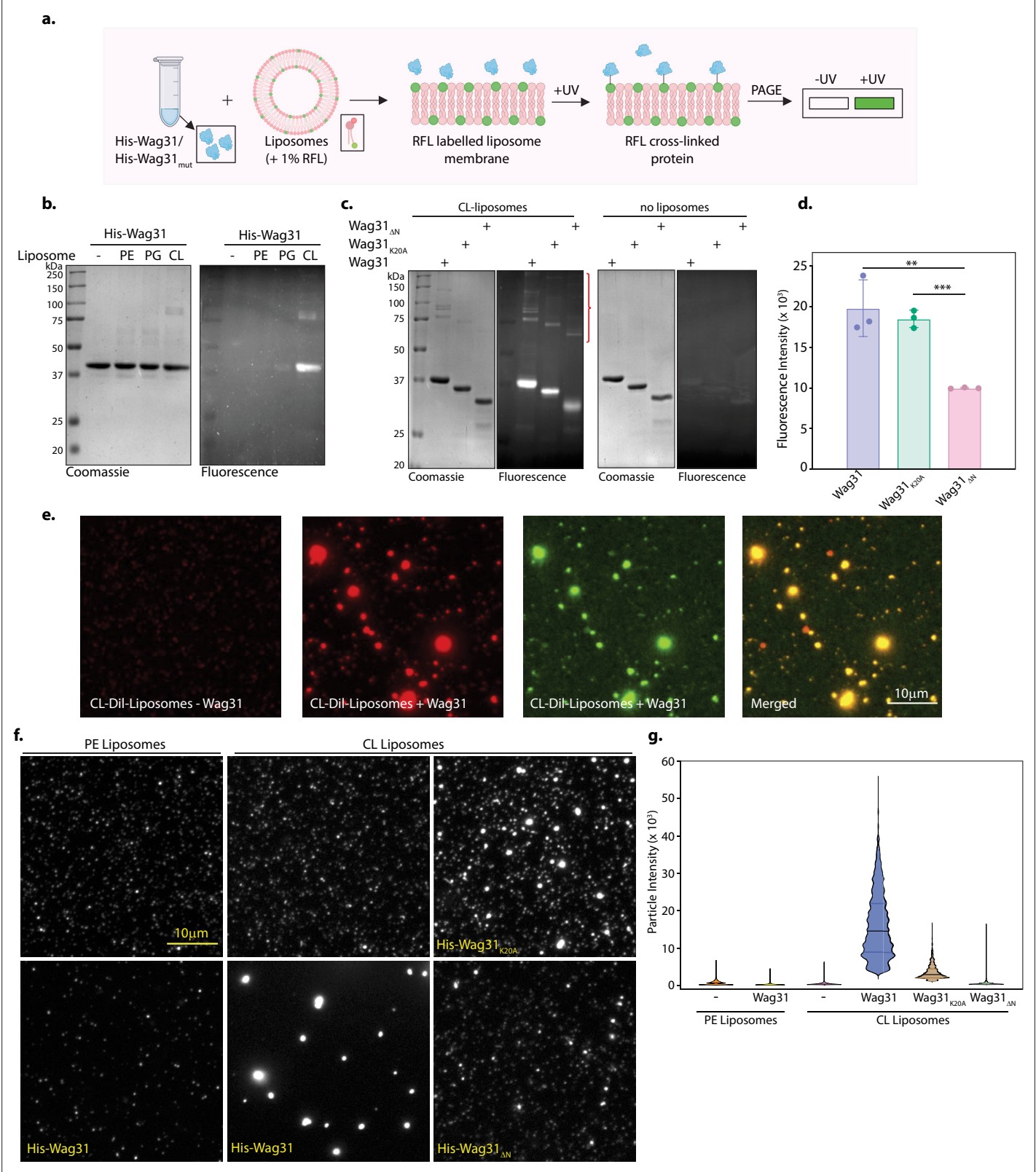

**Figure 4.** Wag31 binds and tethers mycobacterial membrane. (**a**) Schematic showing the workflow of the proximity-based labelling of membrane-associated protein (PLiMAP) assay. Created in BioRender.com. (**b**) Coomassie Brilliant blue (CBB) staining of His-Wag31 and in-gel fluorescence (FI) from a representative PLiMAP experiment performed using 2 μM Wag31 and hundred-fold excess of 30 mol% of either 1,2-dioleoyl-sn-glycero-3-phosphoethanolamine (DOPE-), and1′,3′-bis[1,2-dioleoyl-sn-glycero-phospho]-glycerol (DOPG-), or CL-containing liposomes. (**c**) CBB and FL gels from

*Figure 4 continued on next page*

*Figure 4 continued*

a representative PLiMAP experiment performed with 2 µM Wag31, Wag31$_{K20A}$, or Wag31$_{\Delta N}$ either in the presence or absence of 30 mol% CL liposomes. Note the presence of SDS-resistant higher-order polymers in gel indicated by the red curly bracket. (**d**) Quantitation of PLiMAP data from (**c**) showing a difference in binding propensities of His-Wag31, His-Wag31$_{K20A}$, and His-Wag31$_{\Delta N}$. Data represent the mean ± SD of three independent experiments. Statistical analysis was performed using a two-tailed unpaired Student's *t*-test in GraphPad Prism 9. Wag31 vs Wag31$_{K20A}$ showed no significant difference (ns, p-value = 0.5667) whereas Wag31 vs Wag31$_{\Delta N}$ (p-value = 0.0081, \*\*) and Wag31$_{K20A}$ vs Wag31$_{\Delta N}$ (p-value = 0.0002, \*\*\*) showed significant difference in binding. (**e**) Representative micrographs from binding and tethering assay showing CL-DiI-liposomes (~100 nm) before (first panel) and after incubation (second panel) with 1 µM His-Wag31-GFP. Membrane and protein fluorescence are rendered in red and green, respectively. The experiment was performed independently twice. (**f**) Representative micrographs from tethering experiments performed with His-Wag31, His-Wag31$_{K20A}$, or His-Wag31$_{\Delta N}$ proteins. The left panels indicate phosphatidylethanolamine (PE) liposomes only (upper panel) and PE liposomes incubated with 1 µM Wag31 (lower panel), the middle panels indicate CL liposomes (upper) or CL liposomes incubated with 1 µM Wag31 (lower panel), the right panel indicates CL liposomes either incubated with His-Wag31$_{K20A}$ (upper) and His-Wag31$_{\Delta N}$ (lower). For visualization, PE- and CL-containing liposomes were doped with trace amounts of 3,3'-dilinoleyloxacarbocyanine perchlorate (FAST DiO) and 4-chlorobenzenesulfonate (FAST DiI), respectively. (**g**) Data from particle analysis showing the maximum intensity of liposomes for all the reactions from two independent experiments.

The online version of this article includes the following source data for figure 4:

**Source data 1.** Source data used for making *Figure 4b, c*.

**Source data 2.** Unmarked and uncropped source data used for making *Figure 4b, c*.

**Source data 3.** Source data used for generating the plots *Figure 4d, g*.

**Source data 4.** Source data, a avi movie used for generating *Figure 4e*.

**Source data 5.** Source data, a zip file containing multiple avi movie files used for generating *Figure 4f*.

Wag31-GFP (*Figure 4e*, remaining panels). The Wag31 tetramer has been modelled as a long rod-shaped protein filament (*Choukate et al., 2020*; *Choukate and Chaudhuri, 2020*) and such clustering of liposomes falls in congruence with the model. It indicates that these long filaments of Wag31 act to tether liposomes. We then systematically evaluated this effect by testing liposomes of different compositions and Wag31 mutants. Again, in the absence of Wag31, both PE- and CL-containing liposomes appeared as faint but distinct fluorescent puncta (*Figure 4f*). The addition of Wag31 caused clustering of CL-containing liposomes into bright foci (*Figure 4f*), which is quantified by a particle intensity analysis (*Figure 4g*). This clustering was caused by membrane binding of Wag31 because PE-containing liposomes that do not bind Wag31 (*Figure 4b*) showed no clustering (*Figure 4f, g*). As expected, Wag31$_{\Delta N}$ was defective in membrane binding (*Figure 4c, d*) and showed a severe tethering capacity defect (*Figure 4f, g*). Surprisingly, Wag31$_{K20A}$, which showed a marginal defect in membrane binding (*Figure 4c, d*), showed a marked defect in tethering capacity (*Figure 4f, g*). Together, these results suggest that membrane binding and membrane tethering are non-overlapping attributes in the structure of Wag31, and the N-terminal DivIVA-domain is indispensable for membrane-tethering activity.

## Tethering is crucial for the survival of mycobacteria

Results presented in *Figure 4c* indicate that Wag31 binds strongly to CL-containing membranes. CL is a negatively charged inverted cone-shaped lipid that has been shown to be concentrated at the sites of negative curvature in the bacterial cell, that is, pole and septum (*Renner and Weibel, 2011*; *Oliver et al., 2014*). Wag31 has been demonstrated previously to localize at sites of negative curvature (*Meniche et al., 2014*). Their similar localization pattern, coupled with Wag31's affinity to bind to CL, led us to examine the effect of this binding in vivo. We hypothesized that modulations in Wag31 levels might alter CL distribution in the cell.

To examine this, we utilized a fluorophore 10-N-Nonyl-acridine orange (NAO) that is widely used to examine the localization of CL in prokaryotes and in the mitochondria of eukaryotes (*Renner and Weibel, 2011*; *Garcia Fernandez et al., 2004*; *Mileykovskaya and Dowhan, 2000*). NAO binds to anionic phospholipids with an affinity constant of $7 \times 10^4$ M$^{-1}$ and to CL with an affinity constant of $2 \times 10^6$ M$^{-1}$ (*Petit et al., 1992*). We stained *Msm* and Wag31 depleted and overexpressing cells with NAO and imaged them using a fluorescence microscope. While the localization pattern of NAO was polar in *Msm*+ATc and *Δwag31*-ATc, there was a redistribution of red fluorescence to the cytosol and along the perimeter of the cell in *Δwag31*+ATc (*Figure 5a, b*). We quantified the distribution of fluorescence using a line profile which yielded peaks of intensity at the poles and a basal level of fluorescence along the lateral cell body for both *Msm*+ATc and *Δwag31*-ATc (*Figure 5b*). Wag31 overexpression also led

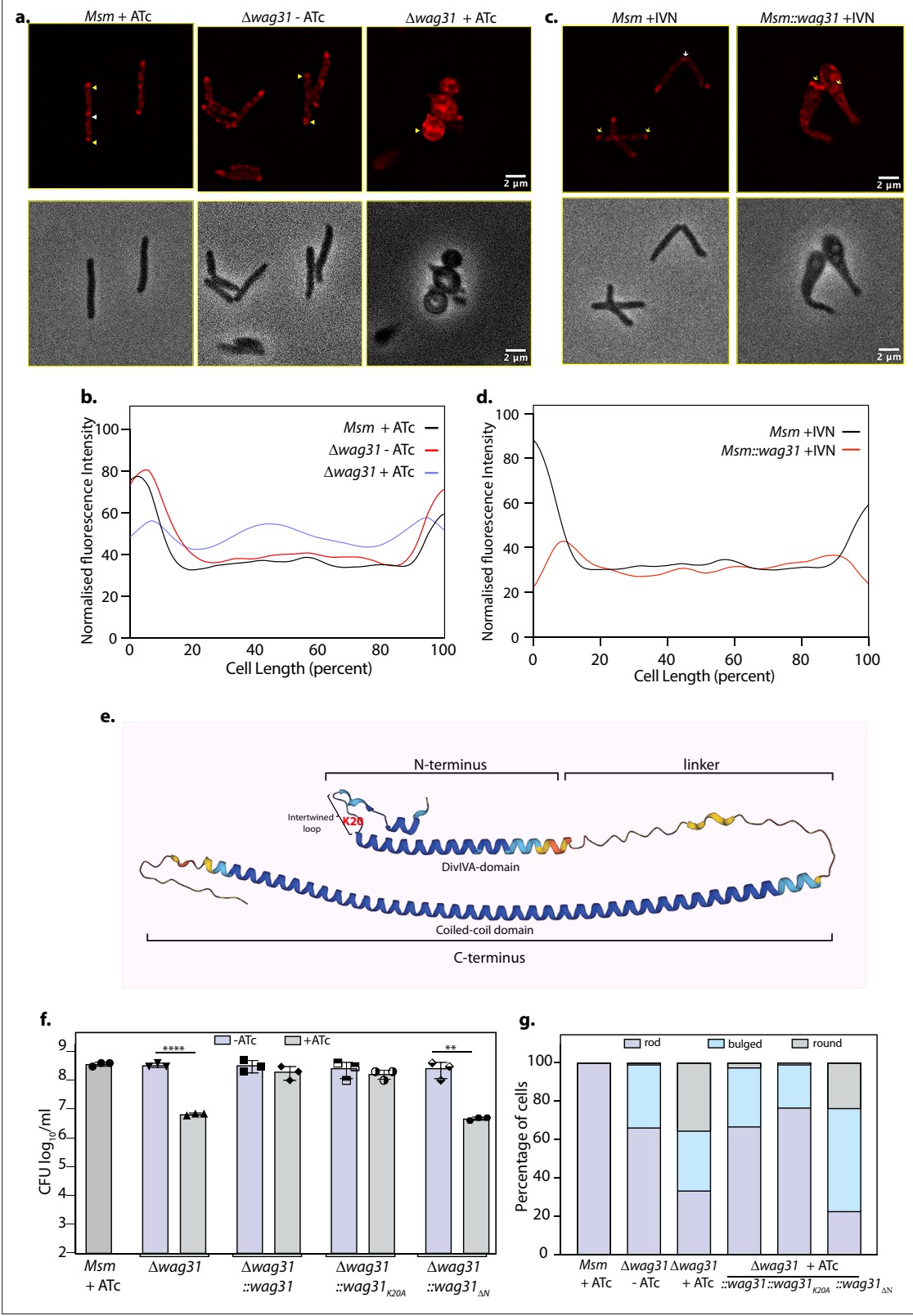

**Figure 5.** Tethering is crucial for the survival of mycobacteria. (**a**) Top panel: Representative fluorescent micrographs showing 10-N-Nonyl-acridine orange (NAO) intensity across *Msm*+ATc, *Δwag31*-ATc, and *Δwag31*+ATc samples imaged at 12 hr post-ATc treatment. Bright red foci of polar CL are indicated with yellow arrowheads and septal CL foci with white arrowheads. Bottom panel: Respective phase images. (**b**) Line plot depicting CL distribution along the length of the cells across *Msm*+ATc, *Δwag31*-ATc, and *Δwag31*+ATc. N = 100 cells across two independent biological replicates

*Figure 5 continued on next page*

*Figure 5 continued*

were analysed using Fiji and their normalized fluorescence intensities were plotted against cell length; fluorescence emitted from 0–20% to 80–100% of cell length was considered polar. (**c**) Representative fluorescent micrographs showing NAO staining in *Msm*+IVN, *Msm::wag31*+IVN samples imaged at 12 hr post 5 µM IVN treatment. Bright red foci of polar CL are indicated with yellow arrowheads, and septal CL foci with white arrowheads. The panels below show corresponding phase images. (**d**) Line plot depicting CL distribution along the length of the cells in *Msm*+IVN, *Msm::wag31*+IVN. N = 100 cells across two independent biological replicates were analysed using Fiji. The *y*-axis represents the normalized fluorescence intensities, and the *x*-axis represents cell length expressed as a percentage of total cell length. Fluorescence peaks in 0–20% and 80–100% of cell length were considered polar. (**e**) Structure of Wag31 (MSMEG_4217) from AlphaFold protein structure database (*Tunyasuvunakool et al., 2021*). The structure represents the N-terminal, linker, and C-terminal domains in Wag31. The N-terminal contains a short helix, followed by an intertwined loop harbouring positively charged amino acid K20 (highlighted in red) and the DivIVA-domain. The N-terminus is linked to the C-terminus via a linker. (**f**) *Msm*+ATc, *Δwag31*, *Δwag31::wag31*, *Δwag31::wag31*K20A, and *Δwag31::wag31*ΔN either untreated or treated with ATc were scored for bacillary survival post 12 hr ATc addition by enumerating CFUs. Appropriate serial dilutions were plated on 7H11 plates (without antibiotic or ATc), and bar graphs demonstrating bacillary survival CFU log₁₀/ml ± standard deviations (SD, represented by error bars) were plotted and statistical significance was calculated using two-way RM ANOVA followed by *Tukey's* multiple comparison test, α = 0.01, GP: 0.1234 (ns), 0.0332 (*), 0.0021 (**), 0.0002 (***),<0.0001 (****). The experiment was performed twice independently, each with triplicates. (**g**) Strains described in (**f**) were scored for their morphology-rod, bulged or round at 12 hr post ATc treatment. 308 cells of each strain across two independent experiments were analysed.

The online version of this article includes the following source data for figure 5:

**Source data 1.** Source data used for making *Figure 5a, b*.

**Source data 2.** Unmarked and uncropped source data used for making *Figure 5a, b.*.

**Source data 3.** Source data used for making *Figure 5b, d, f, g*.

to a change in the distribution of NAO from the poles and septum to 'donut-like' fluorescent foci in the cytosol (*Figure 5c, d*). As the affinity of NAO for CL is more than other anionic phospholipids and our result (shown in *Figure 4b*) indicate that Wag31 prefers to bind CL over PG (another negatively charged phospholipid), the change in the distribution of NAO fluorescence implies that the loss or overexpression of Wag31 causes the mislocalization of CL in mycobacterium.

Subsequently, we investigated the impact of tethering-deficient Wag31 mutants on cell survival. Wag31 has an N-terminal domain joined to the C-terminal via a linker region. The N-terminal domain contains a positively charged patch in the intertwined region (the region between two helices in the N-terminus) and membrane binding has been attributed to this region (*Choukate and Chaudhuri, 2020*). This patch is lined with conserved arginine and lysine residues, K15, K20, and R21 (*Choukate and Chaudhuri, 2020*; *Figure 5e*). As our in vitro results suggest the importance of K20 and the DivIVA-domain in tethering, we performed complementation assays with the tethering-defective mutants to examine cell survival and morphology. *Δwag31* was complemented with an integrative copy of either Wag31 or Wag31K20A or Wag31ΔN.*Δwag31*+ATc showed a two log-fold difference in survival and a 40% decrease in the percentage of rod-shaped cells compared with *Δwag31*-ATc (*Figure 5f, g*). Complementation with Wag31 and Wag31K20A rescued the loss of cell viability and restored the rod shape of the cells (*Figure 5f, g*), whereas Wag31ΔN behaved similar to *Δwag31*+ATc. It neither restored the rod shape of the cells nor reversed the survival defect (*Figure 5f, g*). These results highlight the importance of the N-terminal DivIVA-domain for sustaining mycobacterial morphology and survival.

## Tethering is correlated to lipid levels and CL localization in mycobacteria

As described above, Wag31-mediated membrane tethering is an absolute for the survival of mycobacteria. Perturbations in lipid homeostasis and mislocalization of CL due to altered Wag31 levels hint at a correlation between its ability to function as a homotypic tether and maintain lipid composition and localization. To derive the mechanism behind it, we performed fluorescence imaging experiments with native and tethering defective mutants of Wag31. To investigate a role of tethering in lipid homeostasis in mycobacteria, we stained wildtype and complementation strains with BODIPY and quantified the fluorescence intensities. We observed comparable staining intensity in *Msm*+ATc and *Δwag31*-ATc (*Figure 6a, b*). *Δwag31*+ATc expectedly had the highest BODIPY staining intensity, indicating higher than normal lipid levels (*Figure 6a, b*). Complementation with Wag31 fully restored the lipid levels to normal, whereas complementation with Wag31K20A could not restore the lipid levels completely (*Figure 6a, b*). Wag31ΔN exhibited higher lipid levels than *Δwag31*-ATc (*Figure 6a, b*). These results indicate that tethering could be one of the mechanisms required for maintaining lipid homeostasis.

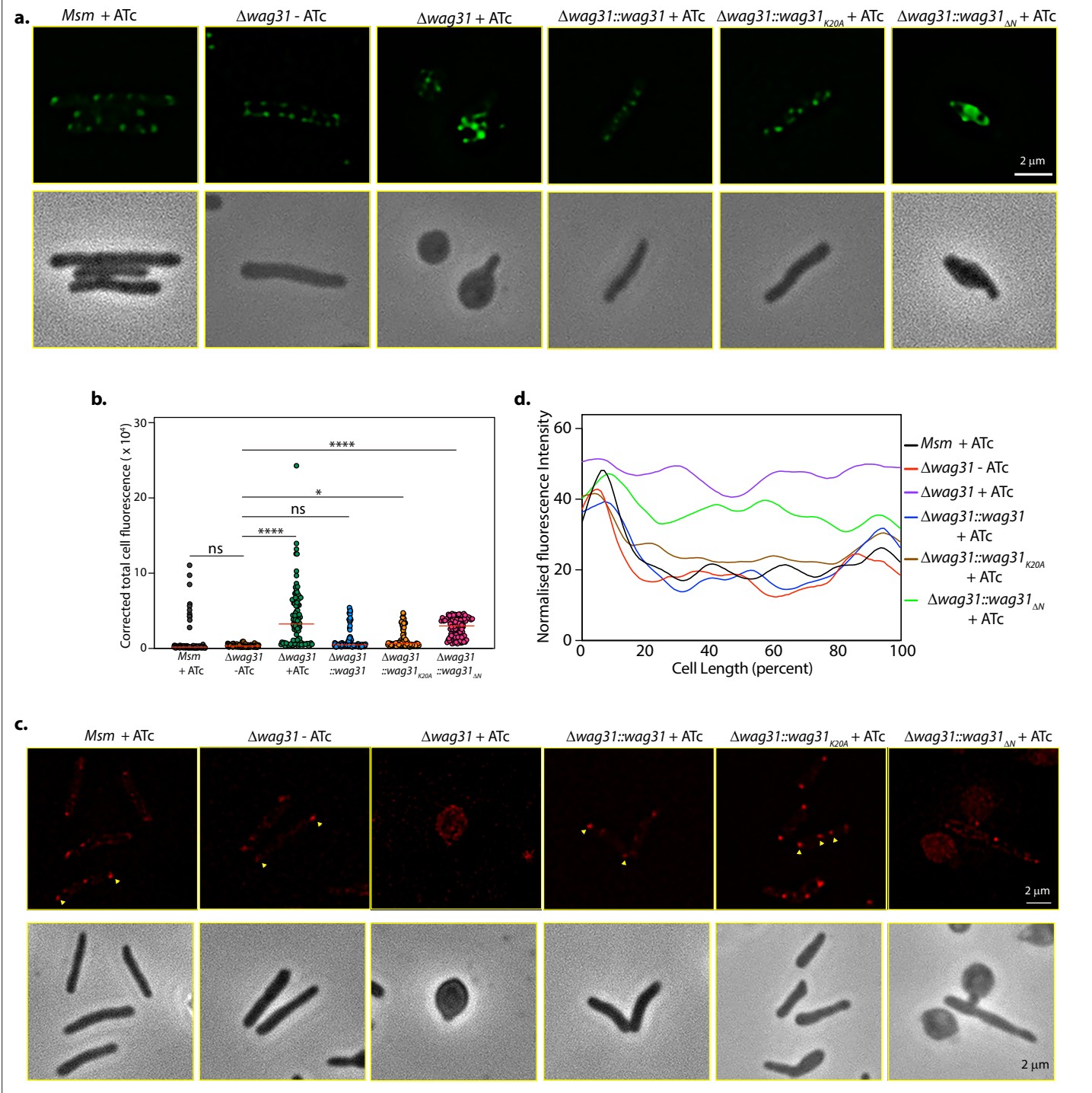

**Figure 6.** Tethering is correlated with lipid levels and CL localization in mycobacteria. (**a**) Top panel: Representative fluorescent micrographs showing BODIPY staining across Δwag31-ATc and Msm, Δwag31, Δwag31::wag31, Δwag31::wag31$_{K20A}$, Δwag31::wag31$_{ΔN}$ imaged at 12 hr post ATc addition. Bottom panel: Corresponding phase images. (**b**) Corrected total cell fluorescence (CTCF) of BODIPY-stained cells was calculated from N = 100 cells across two independent experiments and analysed using Fiji and plotted in a graph. The horizontal bar represents the median CTCF (statistical analysis was performed using one-way ANOVA followed by Brown–Forsythe test (Tukey's multiple comparison test; 0.1234 (ns), 0.0332 (*),<0.0001 (****))). (**c**) Fluorescent micrographs showing 10-N-Nonyl-acridine orange (NAO) intensity of strains as mentioned above. Bottom panel: Corresponding phase images. (**d**) Line plot analysis of strains described in (**c**) depicting CL localization. N = 100 cells across two independent experiments were analysed using Fiji and their normalized fluorescence intensities were plotted against cell length.

The online version of this article includes the following source data for figure 6:

*Figure 6 continued on next page*

*Figure 6 continued*

**Source data 1.** Source data used for generating *Figure 6a, c*.

**Source data 2.** Unmarked and uncropped source data used for generating *Figure 6a, c*.

**Source data 3.** Source data used for generating *Figure 6b, d*.

Next, we examined the dependence of CL localization on tethering by staining *Msm*+ATc, *Δwag31* -ATc, *Δwag31*+ATc, *Δwag31::wag31*+ATc, *Δwag31::wag31*$_{K20A}$+ATc, and *Δwag31:: wag31*$_{ΔN}$+ATc with NAO. NAO staining showed a polar and septal (in dividing cells) localization pattern in the case of *Msm* and *Δwag31*-ATc, whereas it stained the cytosol and perimeter of the Wag31 depleted cells (*Figure 6c, d*). Complementation with Wag31 restored NAO localization completely, whereas Wag31$_{K20A}$ restored it partially, which could be attributed to its limited tethering activity (*Figure 6c, d*). Importantly, we observed a mis-localized pattern of NAO with Wag31$_{ΔN}$ similar to Wag31 depletion (*Figure 6c, d*). These results inform of the dependence of CL localization on membrane tethering activity of Wag31. Taken together, the data suggest the importance of the N-terminal region of Wag31 in regulating lipid homeostasis and localization.

## The N- and C-terminal domains of Wag31 have distinct functions

Wag31 is a 272 amino-acids long protein that contains an N- and a C-terminal domain that are joined by a linker region (*Figure 7a*). In this report, we established a role for the N-terminal region of Wag31 that comprises DivIVA-domain in membrane-tethering (*Figure 4*). We also identified the protein-interaction network of Wag31 with the help of MS/MS analysis (*Figure 3*; *Figure 3—figure supplement 2*). To investigate which domain of Wag31 promotes its interactions with other proteins, we generated *Msm* strains expressing either N-terminal (Wag31$_{ΔN}$) or C-terminal truncated (Wag31$_{ΔC}$) Wag31 (*Figure 7a*) and performed a series of His pulldown experiments with both the mutants. To investigate the protein-binding ability of the domain truncation mutants, we tested them against novel interactors of Wag31 viz. MurG, SepIVA, and Msm2092 from our interactome database (*Figure 3a*) and AccA3, a known interactor of Wag31. MmpS5 served as a non-interactor negative control in the experiment.

We performed His pulldown assays utilizing *E. coli* lysates expressing either Wag31 or Wag31ΔC or Wag31ΔN (*Figure 7b, e*). These lysates were mixed and incubated with *Msm* lysates expressing 3x-FLAG tagged versions of the above-mentioned proteins (*Figure 7c, f*) in separate reactions and subsequently pulled-down with Cobalt beads. The western blot analysis revealed that all the tested interactors bound to Wag31 (*Figure 7d, g*). Expectedly, MmpS5, being a non-interactor of Wag31 did not bind to it (*Figure 7d*). As shown in *Figure 7d*, none of the interactors bound to Wag31$_{ΔC}$, indicating that the presence of N-terminal DivIVA-domain is not sufficient to enable Wag31 to interact with other proteins. Interestingly, interactions with all the proteins were retained in the case of Wag31ΔN (*Figure 7g*), thus suggesting that Wag31 interacts with other proteins via its C-terminal. Taken together, the results suggest that the N-terminal region of Wag31 is involved in membrane tethering through interactions with CL, and the C-terminal region is involved in modulating interactions with other membrane-associated proteins. Thus, N-terminal and C-terminal of Wag31 have distinct molecular functions that together are crucial for maintaining cell shape, lipid homeostasis, and survival of mycobacteria.

## Discussion

Wag31 modulates mycobacterial cell division by acting as an adaptor protein that recruits the elongation machinery to the old pole for unipolar growth of cells (*Kang et al., 2008*; *Nguyen et al., 2007*; *Meniche et al., 2014*). In the absence of Wag31, rod-shaped cells advance into a spheroplast-like stage, which is thought to arise from dysregulated PG metabolism (*Kang et al., 2008*; *Jani et al., 2010*; *Meniche et al., 2014*). Recently, Wag31 has been implicated in maintaining the lipid-rich IMD compartment (*García-Heredia et al., 2021*). But the molecular mechanisms by which Wag31 manages these functions has remained unclear. In this study, we discover an additional function of Wag31 as a membrane tether, which likely reconciles many of the previously ascribed functions of this protein into a unified model. To gain insights into defects associated with a change in cell shape, we imaged

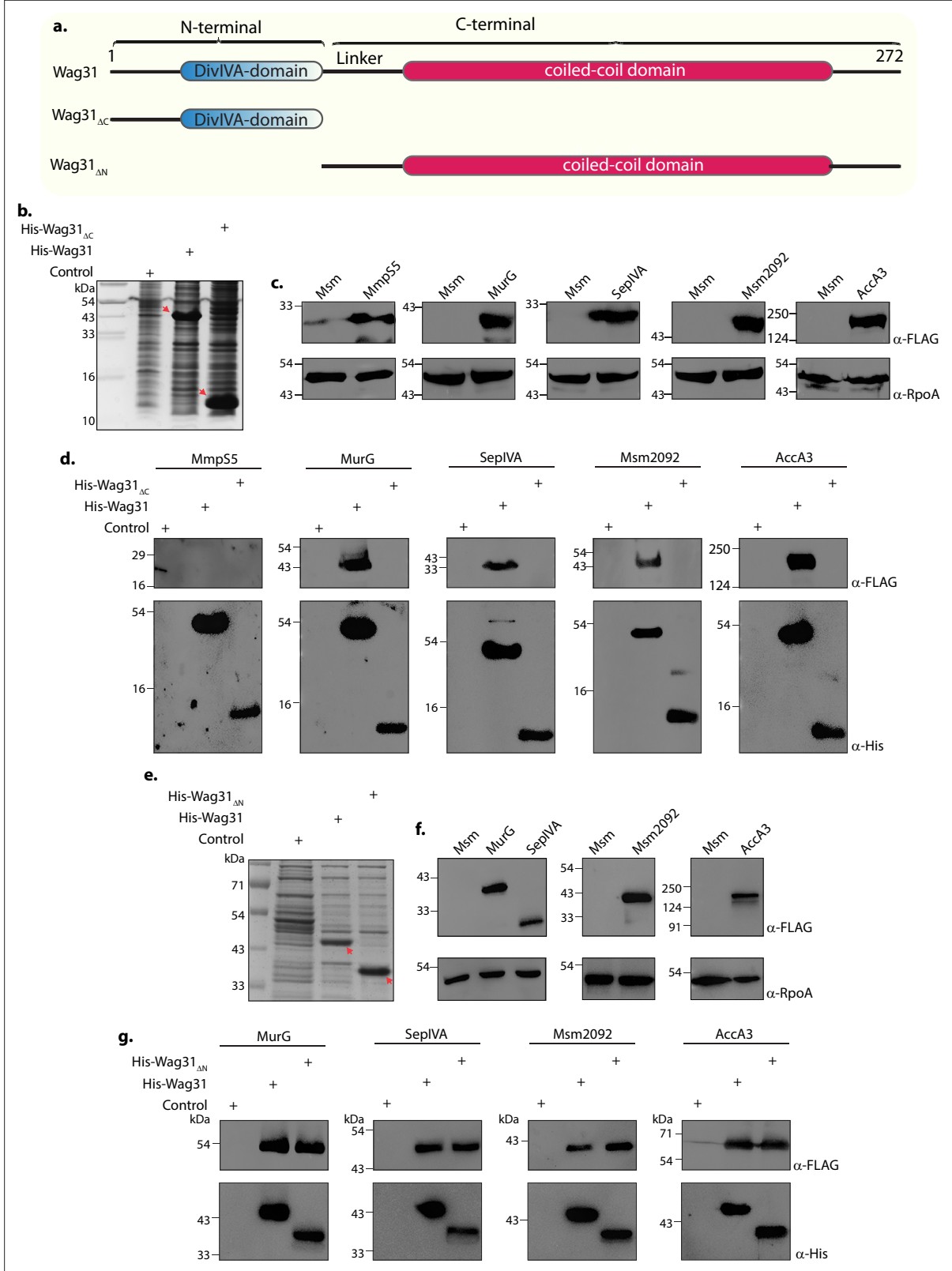

**Figure 7.** The N- and C-terminal domains of Wag31 have distinct functions. (**a**) Illustration depicting the domain architecture of Wag31, Wag31$_{\Delta N}$, and Wag31$_{\Delta C}$. (**b**) Coomassie Brilliant blue (CBB) staining of plain *E. coli* lysate and *E. coli* lysate expressing His-Wag31$_{\Delta C}$. *E. coli* lysate prepared from BL21-DE3 strain was used as the control. (**c**) Representative western blots demonstrating expression of 3XFLAG tagged-MmpS5, -MurG, -SepIVA, -Msm2092, and -AccA3. Whole-cell lysates (WCLs) were prepared from 0.5 µM IVN induced *Msm::mmpS5, Msm::murG, Msm::sepIVA, Msm::msm2092, Msm::accA3*

*Figure 7 continued on next page*

*Figure 7 continued*

and 40 µg of each was resolved a 10% SDS–PAGE, transferred onto a nitrocellulose membrane and probed with α-FLAG and α-RpoA (loading control). (**d**) Representative western blots showing interaction of MurG, SepIVA, Msm2092, and Acca3 with Wag31 and Wag31$_{\Delta c}$. The non-interactor MmpS5 was used as a negative control. *E. coli* and *Msm* lysates represented in (b, c) incubated together and pulled down with Cobalt beads were resolved on Tris Tricine gel as described elsewhere (*Schägger, 2006*), transferred onto nitrocellulose membrane and probed with α-His to detect the pulldown and α-FLAG to detect the interaction. The experiment was performed independently twice. (**e**) Coomassie Brilliant blue (CBB) staining of *E. coli* lysates expressing His-Wag31$_{\Delta N}$. *E. coli* lysate prepared from BL21-DE3 strain was used as the control. (**f**) Representative western blots demonstrating expression of 3XFLAG tagged-MurG, -SepIVA, -Msm2092, and -AccA3. WCLs were prepared from 0.5 µM IVN induced *Msm::murG*, *Msm::sepIVA*, *Msm::msm2092*, *Msm::accA3* and 40 µg of each was resolved a 10% SDS–PAGE, transferred onto a nitrocellulose membrane and probed with α-FLAG and α-RpoA (loading control). (**g**) Representative western blots showing interaction of MurG, SepIVA, Msm2092, and Acca3 with Wag31 and Wag31$_{\Delta N}$. *E. coli* and *Msm* lysates represented in (**e, f**) incubated together and pulled down with Cobalt beads were resolved on 10% SDS–PAGE, transferred onto nitrocellulose membrane and probed with α-His to detect the pulldown and α-FLAG to detect the interaction. The experiment was performed independently twice.

The online version of this article includes the following source data for figure 7:

**Source data 1.** Source data used for generating *Figure 7b–g*.

**Source data 2.** Unmarked and uncropped source data used for generating *Figure 7b–g*.

Wag31-deficient cells and found a dramatic accumulation of lipids in the form of ILIs. Interestingly, both *Msm* and *ΔWag31*-ATc showed the presence of ILIs, but in much less abundance, indicating that ILIs are a normal occurrence that may function to regulate cellular functions (*Figure 1g, h*).

The loss of Wag31 leads to the accumulation of neutral lipids while its overexpression increase the levels of α-MA, implying that the maintenance of native Wag31 levels is critical for the proper balance of lipid composition (*Figure 2f–h*). The lipid classes that get impacted by the depletion of Wag31 vs overexpression are different. Wag31 is an adaptor protein that interacts with proteins of the ACCase complex (*Meniche et al., 2014*; *Xu et al., 2014*) that synthesize fatty acid precursors and regulate their activity (*Habibi Arejan et al., 2022*). The varied response on lipid homeostasis could be attributed to a change in the stoichiometry of these interactions of Wag31. While Wag31 depletion would prevent such interactions from occurring and might affect lipid synthesis that directly depends on Wag31–protein partner interactions, its overexpression would lead to promiscuous interactions and a change in the stoichiometry of native interactions that would ultimately modulate lipid synthesis pathways.

A protein's interactome is akin to its functional fingerprint. Here, we identify 116 novel interacting partners of Wag31 through mass spectrometry and validate three of these interactions, namely with MurG, SepIVA, and Msm2092, through pulldown studies (*Figures 3 and 5i*). However, it can't be ascertained whether interactions are direct or mediated by other proteins. Strikingly, the bulk of Wag31 interactors are membrane proteins, which might be attributed to the formation of Wag31 oligomers on the membrane. Our results identify MurG, SepIVA, which regulates MurG (*Freeman et al., 2023*), the D-aminopeptidase Msm2092 and AmiB, both of which are PG remodellers, as Wag31 interactors. Together, these results contribute towards our understanding of how Wag31 affects PG synthesis (*Figure 3a, d*). AccA3 is an interactor of Wag31 (*Meniche et al., 2014*; *Habibi Arejan et al., 2022*) and catalyses the first committed step in the synthesis of long-chain fatty acids by forming malonyl-CoA. In addition, long-chain-specific acyl dehydrogenase (LCAD), which we report here to be another interactor of Wag31, catalyses the first step in the β-oxidation of fatty acids, which might explain the accumulation of long-chain PEs and long-chain fatty acids.

As mentioned above, the overexpression of Wag31 leads to an upregulation in α-MA levels (*Figure 2h*). The localization of the α-MA transporter MmpL3 depends on polar PG levels (*Carel et al., 2014*; *Arejan et al., 2024*). MurG is an essential enzyme for PG synthesis (*Salmond et al., 1980*) and we report here that it strongly interacts with Wag31 (*Figure 3a, d*). It is tempting to hypothesize that homeostatic control on lipid composition arises from the ability of Wag31 to modulate the activity of enzymes involved in lipid metabolism and PG synthesis. That both depletion and overexpression of Wag31 affect lipid composition is likely also a manifestation of its ability to function as a homotypic tether, where the tethering ability is inherently sensitive to the tether concentration.

Our results are the first to establish a membrane tethering function of Wag31. DivIVA proteins possess an N-terminal membrane-binding domain and a C-terminal coiled-coil domain (*Choukate and Chaudhuri, 2020*; *Lenarcic et al., 2009*). Our data show that disruption of the N-terminal domain

(Wag31$_{\Delta N}$) markedly reduces membrane binding and completely abolishes membrane tethering (*Figure 4f, g*). The N-terminal domain binds CL while the C-terminal coiled-coil domain facilitates multimerization, which together could explain the molecular basis for Wag31 to function in homo-typic tethering of CL-containing membranes. Results from testing mutants defective in membrane tethering indicate that this function is important for the proper localization of CL and eventually the survival of mycobacteria (*Figures 5f and 6c, d*). Thus, alterations in the native levels of Wag31 results in a displacement of CL from the poles (*Figure 5a–d*). However, given NAO's ability to bind to anionic phospholipids (albeit with lower affinity), a smaller degree of NAO redistribution intensity might be coming indirectly from other anionic phospholipids displaced from the membrane due to the loss of membrane integrity or cell shape changes caused by Wag31 depletion.

While, complementation with Wag31 reverses the phenotypic changes observed upon Wag31 depletion, complementation with the membrane tethering-defective mutant (Wag31$_{\Delta N}$) fails to do so (*Figures 5f, g and 6a–d*). These results not only highlight the dependence of CL localization on membrane tethering but also underscores the importance of tethering for maintaining cellular morphology, physiology and survival. CL micro-domains at the poles *Renner and Weibel, 2011*; *Oliver et al., 2014* have been shown to be an important hub for protein localization and activity. Several studies have implicated CL in modulating the activities of membrane-resident proteins involved in electron-transport, protein translocation, bacterial two-component system, etc. (*Carranza et al., 2017*; *Gold et al., 2010*; *Yeo et al., 2023*; *Ouellette et al., 2022*). Our work delineates the roles of the N- and C-terminal domains of Wag31. The N-terminal DivIVA-domain facilitates lipid binding and membrane tethering, and the C-terminal coiled-coil domain engages in protein–protein interactions and facilitates the multimerization of Wag31. This is evident from findings that deletion of the N-terminal DivIVA-domain affects lipid binding and tethering but not its ability to bind interacting partners (*Figures 4c, d, f, g and 7c*). Based on our results, we propose a model in which the N-terminal of Wag31 binds and tethers CL ensuring its proper localization while the C-terminal is free to interact and recruit proteins to the membrane to facilitate cell growth and division (*Figure 8*). We reason that the loss of cell shape and viability in the absence of tethering occurs due to mis-localization of CL which in turn has a negative impact on membrane-localized processes at a global scale.

# Materials and methods

## Key resources table

| Reagent type (species) or resource | Designation | Source or reference | Identifiers | Additional information |
|---|---|---|---|---|
| Genetic reagent | pJV53 | *van Kessel and Hatfull, 2007* | pJV53 plasmid from Dr. Hatfull's lab wherein the kan$^r$ antibiotic cassette was swapped with apra$^r$ cassette. | apra$^{res}$ |
| Genetic reagent | pSTKiT$_{off}$ | *Soni et al., 2015* | L5 integrative pST-KiT plasmid harboring reverse tetR gene. | kan$^{res}$ |
| Genetic reagent | pST-wag31 | | pST-KiT$_{off}$ construct harboring wag31 between NdeI-HindIII sites. | kan$^{res}$ |
| Genetic reagent | pST-gfp | | pST-KiT$_{off}$ construct harboring gfp between NdeI-HindIII sites. | kan$^{res}$ |
| Genetic reagent | pET-wag31 | | pET28b harboring His-wag31-His | kan$^{res}$ |
| Genetic reagent | pET-wag31$_{K20A}$ | | pET28b harboring His-wag31$_{K20A}$-His | kan$^{res}$ |
| Genetic reagent | pET-wag31$_{\Delta N}$ | | pET28b harboring His-wag31$_{\Delta 1-60}$-His | kan$^{res}$ |
| Genetic reagent | pET-wag31-gfp | | pET28b harboring His-wag31-gfp-His | kan$^{res}$ |

*Continued on next page*

*Continued*

| Reagent type (species) or resource | Designation | Source or reference | Identifiers | Additional information |
|---|---|---|---|---|
| Genetic reagent | pSCG-wag31 | | Giles integrative plasmid harboring His-wag31-His under the control of wag31 native promoter | apra$^{res}$ |
| Genetic reagent | pSCG-wag31$_{K20A}$ | | Giles integrative plasmid harboring His-wag31$_{K20A}$-His under the control of wag31 native promoter | apra$^{res}$ |
| Genetic reagent | pSCG-wag31$_{\Delta N}$ | | Giles integrative plasmid harboring His- wag31$_{\Delta N}$-His under the control of wag31 native promoter | apra$^{res}$ |
| Genetic reagent | pNit-wag31 | | IVN inducible plasmid harboring 3X-FLAG-wag31 | kan$^{res}$ |
| Genetic reagent | pNit-murG | | IVN inducible plasmid harboring 3X-FLAG-murG | kan$^{res}$ |
| Genetic reagent | pNit-sepIVA | | IVN inducible plasmid harboring 3X-FLAG-sepIVA | kan$^{res}$ |
| Genetic reagent | pNit-msm2092 | | IVN inducible plasmid harboring 3X-FLAG-msm2092 | kan$^{res}$ |
| Genetic reagent | pNit-accA3 | | IVN inducible plasmid harboring 3X-FLAG-accA3 | kan$^{res}$ |
| Genetic reagent | pNit-mmpS5 | | IVN inducible plasmid harboring 3X-FLAG-mmpS5 | kan$^{res}$ |
| Genetic reagent | pNit-mmpL4 | | IVN inducible plasmid harboring 3X-FLAG-mmpL4 | kan$^{res}$ |
| Gene (*Mycobacterium smegmatis*) | wag31 | Mycobrowser | Uniprot A0R006 | |
| Gene (*M. smegmatis*) | murG | Mycobrowser | Uniprot A0R016 | |
| Gene (*M. smegmatis*) | sepIVA | Mycobrowser | Uniprot A0QV18 | |
| Gene (*M. smegmatis*) | Msm2092 | Mycobrowser | Uniprot A0QU64 | |
| Gene (*M. smegmatis*) | accA3 | Mycobrowser | Uniprot A0QTE1 | |
| Gene (*M. smegmatis*) | mmpS5 | Mycobrowser | Uniprot A0QP03 | |
| Gene (*M. smegmatis*) | mmpL4 | Mycobrowser | Uniprot A0QY12 | |
| strain, strain background (*M. smegmatis*) | Δwag31 | This study | | Hyg$^{res}$, kan$^{res}$, |
| Strain, strain background (*M. smegmatis*) | Msm::gfp | This study | | kan$^{res}$, |
| Strain, strain background (*M. smegmatis*) | Δwag31::wag31 | This study | | Hyg$^{res}$, kan$^{res}$, apramycin$^{res}$ |
| Strain, strain background (*M. smegmatis*) | Δwag31::wag31$_{K20A}$ | This study | | Hyg$^{res}$, kan$^{res}$, apramycin$^{res}$ |
| Strain, strain background (*M. smegmatis*) | Δwag31::wag31$_{\Delta N}$ | This study | | Hyg$^{res}$, kan$^{res}$, apramycin$^{res}$ |
| Strain, strain background (*M. smegmatis*) | Δwag31::wag31$_{\Delta C}$ | This study | | Hyg$^{res}$, kan$^{res}$, apramycin$^{res}$ |
| Strain, strain background (*M. smegmatis*) | Msm::wag31 | This study | | Kan$^{res}$ |

*Continued on next page*

*Continued*

| Reagent type (species) or resource | Designation | Source or reference | Identifiers | Additional information |
|---|---|---|---|---|
| Strain, strain background (*M. smegmatis*) | Msm::murG | This study | | Kan^res |
| Strain, strain background (*M. smegmatis*) | Msm::sepIVA | This study | | Kan^res |
| Strain, strain background (*M. smegmatis*) | Msm::msm2092 | This study | | Kan^res |
| Strain, strain background (*M. smegmatis*) | Msm::accA3 | This study | | Kan^res |
| Strain, strain background (*M. smegmatis*) | Msm::mmpS5 | This study | | Kan^res |
| Strain, strain background (*M. smegmatis*) | Msm::mmpL4 | This study | | Kan^res |
| Genetic reagent (*E. coli*) | *E. coli* DH5α | Invitrogen | | Used for clonings |
| Genetic reagent (*E. coli*) | *E. coli* BL21 DE3 | Novagen | | Used for protein expression |
| Chemical compound | BODIPY | Invitrogen | D3922 | Fluorescent stain |
| Chemical compound | Acridine orange 10-Nonyl bromide (NAO) | Invitrogen | A1372 | Fluorescent stain |
| Chemical compound | 2-dioleoyl-sn-glycero-3-phosphocholine (DOPC) | Avanti polar lipids | 850375 C | Lipid used for PLiMAP |
| Chemical compound | 2-dioleoyl-sn-glycero-3-phosphoethanolamine (DOPE) | Avanti polar lipids | 850725 C | Lipid used for PLiMAP |
| Chemical compound | 1',3'-bis(1,2-dioleoyl-sn-glycero-3-phospho)-glycerol (DOPG) | Avanti polar lipids | 840475 C | Lipid used for PLiMAP |
| Chemical compound | FAST DiI | Invitrogen | D7756 | Lipid used for tethering assay |
| Chemical compound | FAST DiO | Invitrogen | D3898 | Lipid used for tethering assay |
| Chemical compound | Cardiolipin | Avanti polar lipids | 841199 C | Lipid used for PLiMAP |
| Chemical compound | Cobalt agarose beads | GoldBio | H-310–5 | Resin used for His-Pull down assays |
| Chemical compound | FLAG M2 Magnetic beads | Sigma | M8823 | Resin used for Immunoprecipitation |
| Chemical compound | TALON Metal affinity resin | Takara Bio | 635653 | Resin used for Protein purification |
| Chemical compound | Ni-NTA Agarose beads | Qiagen | 30210 | Resin used for Protein purification |

Mass spectrometry samples were processed and performed by Valerian Chem.

## Bacterial strains and culturing

Middlebrook 7H9 medium (BD Biosciences) or 7H11 agar (BD Biosciences) was used to cultivate mycobacterial strains. 7H9 medium was supplemented with 10% ADC (albumin, dextrose, NaCl, and catalase) and 0.2% glycerol (Sigma), and 0.05% Tween-80 (Sigma) or 7H11 agar (BD Biosciences) with 10% OADC (oleic acid added to ADC) and 0.2% glycerol were added to the media. Antibiotic concentrations used: Hygromycin (Hyg) was used at 100 µg/ml, Kanamycin (Kan) at 25 µg/mL, and Apramycin (Apra) at 30 µg/ml for *Msm. E. coli* was cultured in Miller Luria Bertani Broth (Himedia) or Miller Luria Bertani Agar. Antibiotic concentrations used: Hyg was used at 150 µg/ml, Kan at 50 µg/ml, Apra at 30 µg/ml, and Ampicillin (Amp) at 100 µg/ml for *E. coli*.

## Generation of *wag31* conditional mutant and in vitro growth kinetics

Recombineering method, as described in *van Kessel and Hatfull, 2007*, was used to generate a Wag31 conditional gene replacement mutant. First, wag31 was cloned under a tet regulatable

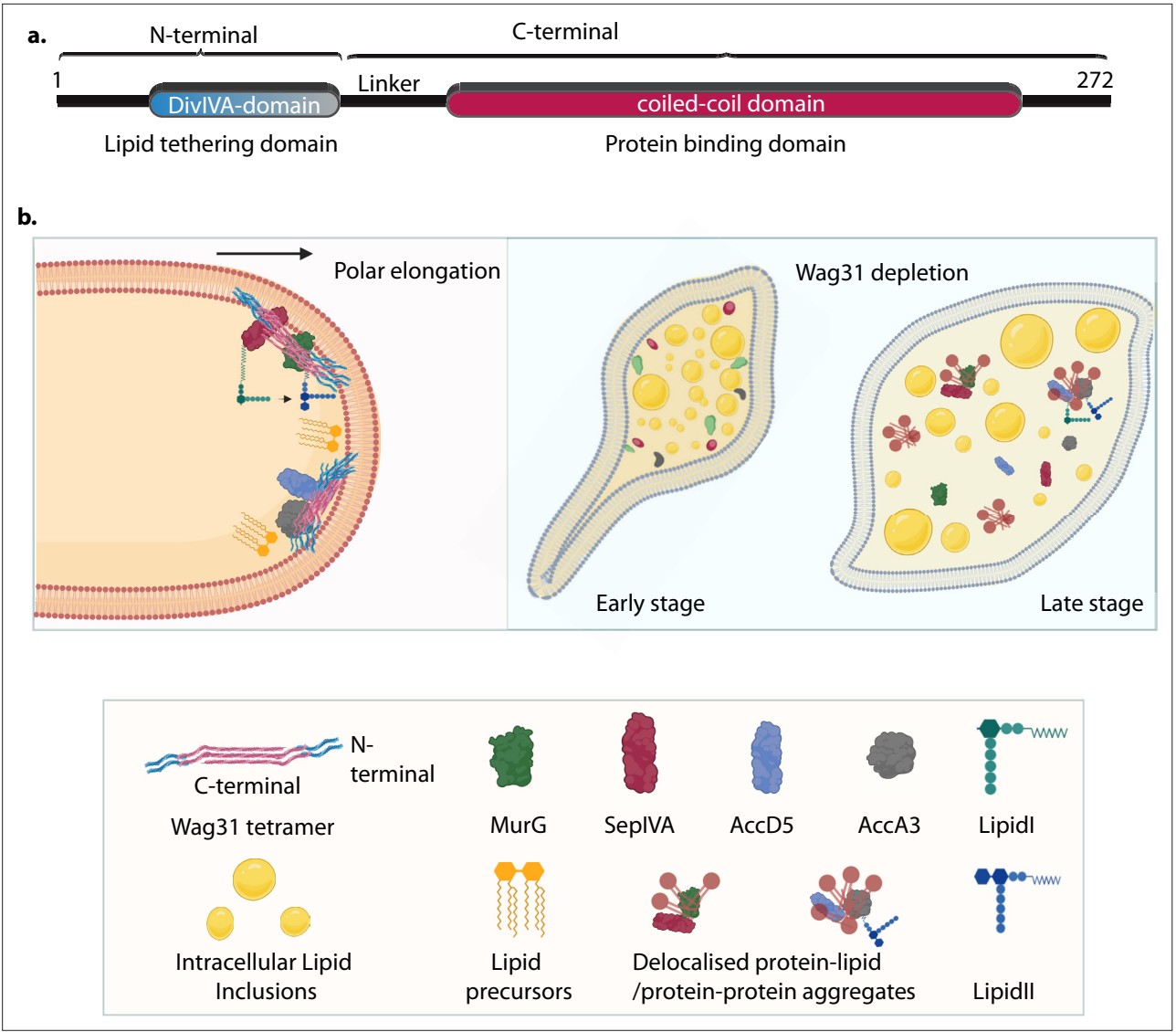

**Figure 8.** Model for Wag31-mediated tethering of the membrane. (**a**) Model of Wag31 showing the N- and C-terminals. N-terminal houses the DivIVA-domain that is involved in membrane tethering and the C-terminal is the protein-binding domain that facilitates interactions of Wag31 with other proteins. (**b**) Left: Illustration showing Wag31 scaffolds bound to cardiolipin in the membrane via N-terminal, tethering it and creating CL microdomains at the poles. The C-terminal of Wag31 bound to protein partners, localize them to the membrane, facilitating several membrane-centric processes such as PG and lipid synthesis, that ensure mycobacterial survival. The right panel shows the time-dependent consequences of Wag31 depletion that is early and late stage. The cells are bulged at one pole and start accumulating intracellular lipid inclusions (ILIs) in the early stage and as the time progresses, the effect of Wag31 depletion becomes severe, rendering the cells round and full of ILIs. In the absence of Wag31, CL microdomains become delocalized due to the loss of tethering, pulling away several membrane-associated proteins causing non-polar growth and loss of curvature. Created in BioRender.com (left, bottom panel). Created in BioRender.com (right panel).

promoter in pST-KiT$_{off}$ (kan res; *Parikh et al., 2013*), which turns off transcription upon the addition of anhydrotetracycline. pST-wag31 construct thus generated was electroporated into a recombineering proficient strain, Msm::pJV53. An allelic exchange substrate was generated by cloning 675 bp from upstream (left-hand sequence, LHS) and 700 bp from downstream (right-hand sequence, RHS) of *wag31* (including ~150 bp at both ends) with *hyg*$^{res}$ cassette and *oriE+cosλ* and digested with SnaBI to release the donor LHS-hyg$^{res}$-RHS. The linear donor was electroporated into Msm::wag31 expressing gp60 and gp61 recombinases. The colonies were selected on Hyg and ten colonies were screened with multiple primer pairs to confirm recombination at the native locus. Confirmed recombinants were passaged multiple times to cure them of pJV53 to generate conditional mutant Δwag31.To perform in vitro growth kinetics, secondary cultures of *Msm* and *Δwag31* were seeded at a density

of $A_{600}$ ~0.05 and either left untreated or treated with 100 ng/ml ATc for 12 hr. At 0 and 12 hr, appropriate serial dilutions were plated on 7H11 agar plate and incubated at 37°C for 2–3 days, and CFUs were enumerated. CFUs were plotted using GraphPad Prism 9 and analysed by performing two-way ANOVA 0.0021 (**), 0.0002 (***).

## Western blot analysis

*Msm*, *Msm::wag31*, and *Δwag31* transformants grown in the presence or absence of IVN/ ATc for 12 hr and harvested by centrifugation at 4000 rpm for 10 min at 4°C. The pellets were resuspended in 1:3 (pellet weight:buffer volume) phosphate buffer saline (PBS) containing 5% glycerol and 1 mM protease inhibitor – PMSF. The resuspension was transferred into pre-equilibrated bead-beating tubes filled to one-fourth with 0.1 mm Zirconia beads. Pre-equilibration was performed by washing the beads with autoclaved MilliQ followed by lysis buffer. The tubes were placed in a bead-beater and subjected to six to eight cycles of bead-beating –1 min on and 2 min rest on ice. To obtain WCL, the tubes were centrifuged at 13,000 rpm for 10 min at 4°C, and the supernatant was collected in a microcentrifuge tube. The concentration of protein in each sample was estimated by Pierce BCA protein Assay kit (Thermo Fischer Scientific), and 40 μg WCLs were resolved on 10–12% SDS–PAGE gels at constant current and transferred onto nitrocellulose membrane. This was followed by blocking the membrane with 5% bovine serum albumin (BSA) for an hour at RT and subsequently incubating it with the desired primary antibody for 2 hr at RT or overnight at 4°C. The primary antibodies used were either anti-RpoA (raised in rabbit), anti-Wag31 (raised in rabbit), anti-FLAG (raised in mice, Sigma), and anti-His (raised in rabbit, CST). The membrane was washed thrice for 10 min each with PBST (PBS + 0.5% Tween-20) to remove any non-specific binding. Next, the membrane was probed with appropriate secondary antibody (anti-rabbit, Invitrogen or anti-mice, Invitrogen) for an hour at RT, washed thrice for 10 min each with PBST, developed using ECL reagent (Millipore), and visualized on Bio-Rad ChemiDoc Touch Gel imaging system.

## Sample preparation for SEM and TEM

*Msm* and *Δwag31* were set up at $A_{600}$~0.05 and grown overnight in either the presence of absence of 100 ng/ml ATc. Post 12 hr, the cells were harvested by centrifugation at 4000 rpm for 10 min at RT. Cells equivalent to $A_{600}$ ~2 were taken further for sample preparation and washed thrice with 5 ml sterilized PBS. The cells were fixed with 5 ml of fixative (4% paraformaldehyde (Sigma), 2.5% glutaraldehyde (EMS), 0.1 M Na-Cacodylate) for 4 hr at RT protected from light. Cells were tap-mixed every 30 min to ensure that they remained in suspension. This was followed by washing the cells twice for 10 min with 0.1 M Na-Cacodylate buffer. Subsequently, cells were stained with 3 ml of 1% Osmium Tetraoxide ($Os_4O_2$) protected from light for a maximum of 1.5 hr with tap-mixing every 30 min. $Os_4O_2$ was removed from the samples by centrifugation at 4000 rpm at 10 min at RT and discarded as per safety guidelines. The cells were then subjected to serial dehydration at RT with increasing concentrations of ethanol from 25% to 100%. Briefly, the cells were incubated for 5 min with 25% ethanol, harvested, and incubated for 7 min with 50% ethanol. This was followed by serially incubating the cells for 15, 20, and 30 min for 75%, 95%, and 100% (thrice) ethanol, respectively. Cells were then taken forward for critical-point drying with the help of 100 μl Hexamethyldisilazane (HMDS). Meanwhile, stubs were prepared by placing carbon tape and coverslip on them. 10 μl of the dried sample was spotted on the coverslip and evenly spread. Next, the stub was coated with 0.8 mm Gold and stored in a desiccator or imaged at ×20,000 using FEI Nova NanoSEM 450. For TEM, cells were grown and harvested as described above for SEM. For TEM sample preparation, 50 OD cells were taken, fixed, stained, and gradually dehydrated with ethanol, as explained above. The cells were then infiltrated with Epon 812 resin. This was followed by cutting the resin into several 62 nm thick ultra-thin sections using an ultramicrotome (Leica EM UC7). The sections were then stained with uranyl acetate and lead citrate and examined on a 120-kV TEM (L120C, Talos, Thermo Scientific) at ×17,500 magnification.

## BODIPY labelling, imaging, and analysis

1 ml cells of *Msm* and *Δwag31* either untreated or treated with 100 ng/ml ATc for 12 hr were stained with 20 μg/ml BODIPY (Invitrogen). For examining lipids in Wag31 overexpression strain, *Msm* and *Msm::wag31* were treated with 5 μM IVN for 12 hr and 1 ml cells of both strains were stained with 2.5 μg BODIPY. Staining was performed for 30 min protected from light in an incubator shaker followed

by washing with sterilized PBS. For Msm and Δwag31, ~10 μl of cells were spotted on a 1% agar pad (prepared in 7H9) and visualized using a 100×/1.4 NA oil immersion objective on a Zeiss, LSM880 at 488/510 nm (excitation:emission). Background correction in the collected images followed by corrected total cell fluorescence analysis was performed using Fiji (*de Boer et al., 1991*) and plotted using GraphPad Prism 9. For examining lipids in BODIPY-stained Msm and Msm::wag31, 2 μl of cells were spotted on a 1% agar pad (prepared in 7H9) and visualized using a 100×/1.32 NA oil immersion objective on a Leica THUNDER Imaging systems at 475:519 nm (excitation:emission). About 20 Z-stack images, each spaced 0.1 μM apart, were acquired using a scientific CMOS K8 camera (Leica microsystems). Background correction and Thunder image processing were performed on the images using the Las-X software module (Leica microsystems). Processed images were then analysed for corrected total cell fluorescence using Fiji (*de Boer et al., 1991*) and plotted using GraphPad Prism 9.

## Sample preparation for Lipidomics and analysis

Secondary cultures of *Msm* and *Δwag31* were seeded at a density of $A_{600}$ ~0.05 and either left untreated or treated with 100 ng/ml ATc for 12 hr until $A_{600}$ ~0.8–1.0. Post 12 hr incubation with ATc, cells corresponding to $A_{600}$ ~20 were harvested and processed for lipid extraction. Bacterial pellets were resuspended in 1 ml sterilized PBS and transferred to a 3 ml glass vial. 2:1 chloroform:methanol mixture was added to it and vortexed vigorously for 2 min. Chloroform contained 1 nm 15:0 FFA to be used as the internal standard for normalization. The samples were centrifuged at 1500 rpm for 5 min at RT to separate the organic layer. The latter was transferred to a new glass vial, and 2 ml chloroform (containing 2% formic acid) was added to the old vial to extract phospholipids from the sample. The samples were centrifuged at 1500 rpm for 5 min at RT to extract the phospholipid-containing organic layer. Both organic layers were pooled and dried under a stream of Nitrogen gas at RT. The dried lipid extracts were re-solubilized in 200 μl of 2:1 $CHCl_3$:$CH_3OH$, and 10 μl was injected into an Agilent 6545 QTOF (quadrupole-time-of-flight) instrument for semi-quantitative analysis using high-resolution auto MS/MS methods and chromatography techniques. A Gemini 5U C-18 column (Phenomenex) coupled with a Gemini guard column (Phenomenex, 4 × 3 mm, Phenomenex security cartridge) was used for LC separation. The solvents used for negative ion mode were buffer A: 95:5 $H_2O$:$CH_3OH$ + 0.1% ammonium hydroxide and buffer B: 60:35:5 isopropanol:$CH_3OH$:$H_2O$ + 0.1% ammonium hydroxide. The 0.1% ammonium hydroxide in each buffer was replaced by 0.1% formic acid + 10 mM ammonium formate for positive ion mode runs. All LC/MS runs were for 60 min, starting with 0.3 ml/min 100% buffer A for 5 min, 0.5 ml/min linear gradient to 100% buffer B over 40 min, 0.5 ml/min 100% buffer B for 10 min, and equilibration with 0.5 ml/min 100% buffer A for 5 min. The following settings were used for the ESI-MS analysis: drying gas and sheath gas temperature: 320°C, drying gas and sheath gas flow rate: 10 l/min, fragment or voltage: 150 V, capillary voltage: 4000 V, nebulizer (ion source gas) pressure: 45 psi, and nozzle voltage: 1000 V. For analysis, a lipid library was employed in the form of a Personal Compound Database Library (PCDL), and the peaks were validated based on relative retention times and fragments obtained. This library was selectively curated from the MycoMass database and the LIPID MAPS Structure Database (LMSD). All robustly detected lipid species were quantified by normalizing areas under the curve (AUC) to the AUC of the relevant internal standard added and by normalizing to the total cell number. $Log_2$ fold changes were then plotted for each lipid compared to the Msm-ATc samples.

## Immunoprecipitation and MS/MS analysis

250 ml secondary cultures of *Msm::gfp* and *Δwag31* were inoculated at an $A_{600}$ ~0.05 and incubated in a shaker incubator until $A_{600}$ ~0.8–1.0. Cells were collected and lysed using bead-beating to make WCL and passed through a 0.2-μM filter. 3 mg lysate from each group was immunoprecipitated overnight with anti-FLAG M2 magnetic beads (Millipore), washed thrice, and heated in 4× Laemmli Buffer. One-fifth of the immunoprecipitation (IP) supernatant was resolved on a 10% gel, transferred onto a nitrocellulose membrane, and probed with anti-FLAG Antibody (Sigma). Lysates from confirmed IPs (biological quadruplets) were taken further for mass spectrometry sample preparation and analysis. Sample preparation and, mass spectrometry analysis were outsourced to VProteomics, New Delhi. Briefly, the IP samples were reduced with 5 mM TCEP (tris(2-carboxyethyl)phosphine), followed by alkylation with 50 mM iodoacetamide. The samples were digested with 1 μg Trypsin for 16 hr at 37°C. Digests were cleaned using a C18 silica cartridge and resuspended in a buffer containing 2%

acetonitrile and 0.1% formic acid. 1 µg of peptide was used for analysis in an Easy-nlc-1000 system coupled to an Orbitrap Exploris mass spectrometer. The gradients were run for 110 min. MS1 spectra were acquired in the Orbitrap (Max IT = 60 ms, AGQ target = 300%; RF Lens = 70%; $R$ = 60 K, mass range = 375–1500; profile data). Dynamic exclusion was employed for 30 s, excluding all charge states for a given precursor. MS2 spectra were collected for the top 20 peptides. MS2 (Max IT = 60 ms, $R$ = 15 K, AGC target 100%). Proteome Discoverer (v2.5) was used to analyse the RAW data against the Uniprot Msm database. The precursor and fragment mass limitations for the dual Sequest and Amanda searches were established at 10 ppm and 0.02 Da, respectively. The false discovery rate for proteins and peptide spectrum matches was adjusted to 0.01 FDR. For identification of interactors unique to Wag31, we removed all the proteins obtained in the four replicates of *Msm::gfp* from the *Δwag31* datasets. In the corrected datasets, proteins present in at least three out of the four replicates with minimum unique peptides #2 were considered interactors. To identify top hits, PSM cut-off was set to 18 and unique peptides to 5.

## Protein expression and purification

BL21(DE3) cells transformed with pET-wag31, or pET-wag31K20A or pET-wag31$_{ΔN}$ were cultured at 37°C until the $A_{600}$ ~0.6. 0.1 mM IPTG (Isopropyl β-D-1-thiogalactopyranoside) was added to the culture to induce protein expression, followed by incubating the culture at 18°C for 12 hr. Cells were pelleted and stored at –40°C. The bacterial pellet was resuspended in a lysis buffer (20 mM HEPES pH 7.4, 150 mM NaCl, and 20 mM imidazole with 1 mM phenylmethylsulphonyl fluoride (PMSF)) and lysed by sonication in an ice-water bath. Lysate was spun at 30,000 × $g$ for 20 min, and the supernatant obtained was incubated with TALON Metal Affinity Resin (Takara Bio) 4°C for 30 min. The resin was extensively washed with lysis buffer, and the bound protein was eluted with 20 mM HEPES pH 7.4, 150 mM NaCl, and 250 mM imidazole. The elution was dialysed against 20 mM HEPES pH 7.4 150 mM NaCl and stored at 4°C for the duration of the experiments. Aggregates were removed by centrifugation at 100,000 × $g$. Concentration of the proteins was quantified by measuring $A_{280}$ using the molar extinction coefficient predicted by the Expasy ProtParam tool.

## Liposome preparation

1,2-Dioleoyl-sn-glycero-3-phosphocholine (DOPC), 1,2-dioleoyl-sn-glycero-3-phosphoethanolamine (DOPE), and 1′,3′-bis[1,2-dioleoyl-sn-glycero-3-phospho]-glycerol (sodium salt), CL were obtained from Avanti Polar Lipids. 1,1′-Dilinoleyl-3,3,3′,3′-tetramethylindocarbocyanine, 4-chlorobenzenesulfonate (FAST DiI), and 3,3′-dilinoleyloxacarbocyanine perchlorate (FAST DiO) were obtained from Invitrogen. The UV-activable, diazirine-containing reactive fluorescent lipid probe BODIPY-diazirine phosphatidylethanolamine (BDPE) was prepared as described earlier (*Bi and Lutkenhaus, 1991*; *de Boer et al., 1991*). For liposome preparation, lipids were aliquoted from chloroform stocks at desired ratios in a clean glass tube and dried under a high vacuum for an hour to a thin film. The dried lipids were used at a final concentration of 1 mM (deionized water was added to them). After hydrating at 50°C for 30 min, lipids were vigorously vortexed and then extruded through polycarbonate filters with pore sizes of 100 nm (Whatman). For PLiMAP assays, liposomes were CL, DOPG, or DOPE with 69 mol% DOPC and 1 mol% BDPE. Liposomes used in tethering assays were made similarly with 30 mol% CL or DOPE and 1 mol% of the fluorescent lipid DiI or DiO in the DOPC background.

## Proximity-based labelling of membrane-associated proteins

For carrying out the PLiMAP assay, Wag31 and its mutants were purified and desired liposomes were prepared as described in supplementary. PLiMAP was carried out and analysed as described (*Jose et al., 2020*). Briefly, 2 µM Wag31 or its mutants were incubated with liposomes containing 30 mol% CL, DOPG, or DOPE (200 µM total lipid) in a final volume of 30 µl. The reaction was incubated for 30 min in the dark at room temperature followed by exposure to a 1-min long pulse of 365 nm UV light (UVP crosslinker CL-1000L) at an intensity of 200 mJ cm$^{-2}$. The reaction was mixed with Laemmli sample buffer, boiled, and equal volumes for all reactions were loaded onto a 10% polyacrylamide gel and resolved using SDS–PAGE. iBright1500 (Thermo Fischer Scientific) or Amersham Typhoon (GE) were used for imaging gels. Gels were first imaged for BODIPY fluorescence and later fixed and stained with Coomassie Brilliant Blue.

### Tethering assays, fluorescence imaging, and image analysis

1 µM of Wag31 and its mutants were mixed with 10 µM of 100 nm extruded liposomes in Eppendorf. The mixture was then transferred to a LabTek chamber and imaged using 100× 1.4 NA oil-immersion objective on an Olympus IX73 inverted microscope connected to an LED light source (CoolLED) and an Evolve 512 EMCCD camera (Photometrics). Image acquisition was controlled by µManager, and images were analysed using Fiji. For single liposome intensity analysis, all images from the same condition from two independent experiments were combined into one montage. The montage was duplicated and thresholded using Fiji's Otsu algorithm to generate a binary mask. The mask was overlaid onto the original montage to obtain the maximum intensity of liposomes.

### NAO labelling, imaging, and analysis

*Msm*, *Msm::wag31* were treated with 5 µM IVN for 12 hr and *Msm* and *Δwag31* were either left untreated or treated with 100 ng/ml ATc for 12 hr. For complementation experiment, 1 ml of each strain was stained with 2 µM NAO for 1 hr protected from light in an incubator shaker and washed thrice with sterilized 1× PBS. 2 µl of cells were spotted on a 1% agar pad (prepared in 7H9) and visualized using a 100×/1.32 NA oil immersion objective on a Leica THUNDER Imaging systems at 475:535 nm (excitation:emission). About 20 Z-stack images, each spaced 0.1 µM apart, were acquired using a scientific CMOS K8 camera (Leica microsystems). For CL-specific labelling, collection was also performed at 642 nm.

Background correction and Thunder image processing were performed on the images using the Las-X software module (Leica microsystems). Processed images were then analysed using Fiji (*Schindelin et al., 2012*) and plotted using GraphPad Prism 9. Briefly, $N = 100$ cells across two independent experiments were examined for their profile of CL distribution by normalizing the cell length and assigning pole with brighter NAO fluorescence intensity as 0 and the other pole as 100. The fluorescence intensity was normalized to the highest fluorescent value for each cell. Corresponding fluorescence intensity value for all cells for every point between 0 and 100 were calculated from the LOOKUP function of excel. The values representing CL fluorescence across 100 cells were averaged out and plotted against normalized cell length in GraphPad prism 9. 0th order smoothening with four neighbours on each size was performed on the curves obtained.

### Complementation experiment

Secondary cultures of *Msm*, *Δwag31*, *Δwag31::wag31*, *Δwag31::wag31*$_{K20A}$, and *Δwag31::wag31*$_{ΔN}$ were set up at $A_{600}$ ~0.05. For each strain, two types of cultures were set up in triplicates – untreated and treated with 100 ng/ml ATc and grown for 12 hr. At 0 and 12 hr, appropriate serial dilutions were plated on 7H11 agar plates and incubated at 37°C for 2–3 days. CFUs were enumerated and plotted using GraphPad Prism 9. Statistical analysis was performed using two-way RM ANOVA followed by *Tukey's* multiple comparison test, $\alpha = 0.01$, GP: 0.1234 (ns), 0.0332 (*), 0.0021 (**), 0.0002 (***), <0.0001 (****). For BODIPY and NAO labelling, the strains were grown as described above. At 12 hr, 1 ml cultures of *Msm*+ATc, *Δwag31*, *Δwag31*+ATc, *Δwag31::wag31*+ATc, *Δwag31::wag31*$_{K20A}$+ATc, and *Δwag31::wag31*$_{ΔN}$ +ATc were taken and labelled with either BODIPY or NAO. The labelling, imaging, and analysis for both fluorophores were performed as described in the respective sections above.

### His pulldown assay

BL21 (DE3) cells transformed with no plasmid, or pET-wag31 or pET-wag31$_{ΔN}$ or pET-wag31$_{ΔC}$ were cultured and cell lysates were prepared as described for PLiMAP and tethering assays. Wag31 or Wag31$_{ΔN}$ was assessed by resolving the samples on 10% SDS-PAGE followed by Coomassie staining whereas Wag31$_{ΔC}$ was resolved on Tris-Tricine PAGE as described elsewhere (*Schägger, 2006*). The interacting partners, namely *murG*, *sepIVA*, *msm2092*, and *accA3* and the non-interactor *mmpS5* were amplified from Msm genomic DNA using gene-specific primers harbouring Ndel-HindIII sites and amplicons were digested and cloned into the corresponding sites in pNit-3X-FLAG shuttle vector. The constructs thus generated were electroporated in *Msm* to generate *Msm::murG*, *Msm::SepIVA*, *Msm::msm2092*, *Msm::accA3*, and *Msm::mmpS5*. The cultures grown till $A_{600}$ ~0.8 were used for fresh inoculation at $A_{600}$ of 0.05 in 7H9 media containing 0.5 µM IVN. Cultures were grown for 12 hr and lysates were prepared by bead-beating and 40 µg were resolved on 10% gels and probed with α-FLAG antibody.

500 µg of *E. coli* lysates expressing Wag31 or mutant were mixed with 300 µg *Msm* lysates expressing Wag31 interactors in a total reaction volume of 700 µl and the samples were twirled at 4°C for 2 hr. 50 µl cobalt beads/sample (GoldBio) were washed twice with lysis buffer and incubated with 5% BSA solution for 2 hr at 4°C. Subsequently, beads were washed with lysis buffer and incubated with samples for 2 hr (twirled at 4°C). The pull downs were washed thrice in lysis buffer containing 1% Triton X-100 and finally resuspended in 2× SDS sample buffer. Pulldown samples were resolved on 10–12% SDS–PAGE and probed with α-His and α-FLAG antibodies.

## Acknowledgements

We thank Dharani Kumar, SEST, University of Hyderabad for helping with imaging of SEM stubs; CCMB for access to its Genomics, Advanced Microscopy, and Electron Microscopy facilities; VIDO, USask for providing infrastructure to carry out some part of the work; Asis Kumar Khuntia for the generation of pET28b-*wag31*$_{K20A}$ and pET28b-*wag31*$_N$. YK was supported by a Research Associateship from NIH (Grant 6044-SC23-26). YK and ND acknowledge MITACS for the Globalink Research Award (FR118569) which supported the work carried out at VIDO. ND also acknowledges support from Canadian Institutes of Health Research (CIHR, funding reference number 185715). VIDO receives operational funding from the Government of Saskatchewan through Innovation Saskatchewan and the Ministry of Agriculture and from the Canada Foundation for Innovation through the Major Science Initiatives Fund. DD thanks DBT for Senior Project Associateship (Grant No. BT/PR13522/COE/34/27/2015). Research in this publication was supported by SERB grant (CRG/2018/001294), Govt. of India, and JC Bose Award (JCB/2019/000015), Govt. of India to VKN. Work in the SSK laboratory was supported by the Swarna Jayanti Fellowship to SSK by SERB, Govt. of India (Grant No. SB/SJF/2021-22/01). AC was supported by a Senior research fellowship from CSIR, India. HK thanks the Indian Institute of Science Education and Research, Pune, for a graduate fellowship. TP thanks the Howard Hughes Medical Institute for funding support.

## Additional information

### Competing interests

Vinay Kumar Nandicoori: Reviewing editor, *eLife*. The other authors declare that no competing interests exist.

### Funding

| Funder | Grant reference number | Author |
|---|---|---|
| Science and Engineering Research Board | CRG/2018/001294 | Vinay Kumar Nandicoori |
| Science and Engineering Research Board | JCB/2019/000015 | Vinay Kumar Nandicoori |
| Mitacs | FR118569 | Yogita Kapoor Neeraj Dhar |
| Howard Hughes Medical Institute | | Thomas John Pucadyil |
| Canadian Institutes of Health Research | 185715 | Neeraj Dhar |
| Science and Engineering Research Board | SB/SJF/2021-22/01 | Siddhesh S Kamat |
| National Institutes of Health | 6044-SC23-26 | Vinay Kumar Nandicoori Yogita Kapoor |
| Department of Biotechnology | BT/PR13522/ COE/34/27/2015 | Vinay Kumar Nandicoori Debatri Dutta |
| CSIR, India | Senior Research Fellowship | Arnab Chakraborty |

| Funder | Grant reference number | Author |
|---|---|---|
| Indian Institute of Science Education and Research Pune | Graduate Fellowship | Himani Khurana |

The funders had no role in study design, data collection, and interpretation, or the decision to submit the work for publication.

## Author contributions

Yogita Kapoor, Conceptualization, Data curation, Formal analysis, Funding acquisition, Validation, Investigation, Visualization, Methodology, Writing – original draft, Writing – review and editing; Himani Khurana, Data curation, Formal analysis, Validation, Investigation, Visualization, Writing – original draft, Writing – review and editing; Debatri Dutta, Formal analysis, Validation, Investigation, Visualization; Arnab Chakraborty, Data curation, Formal analysis, Validation, Investigation, Writing – original draft, Writing – review and editing; Anshu Priya, Investigation; Archana Singh, Methodology; Siddhesh S Kamat, Supervision, Validation, Visualization, Writing – review and editing; Neeraj Dhar, Conceptualization, Resources, Supervision, Funding acquisition, Validation, Investigation, Visualization, Methodology, Writing – original draft, Writing – review and editing; Thomas John Pucadyil, Conceptualization, Supervision, Validation, Visualization, Writing – original draft, Writing – review and editing; Vinay Kumar Nandicoori, Supervision, Funding acquisition, Visualization, Writing – original draft, Project administration, Writing – review and editing

## Author ORCIDs

Yogita Kapoor ⓘ https://orcid.org/0000-0001-8543-1755
Arnab Chakraborty ⓘ https://orcid.org/0000-0001-5812-3984
Anshu Priya ⓘ https://orcid.org/0009-0007-0057-3174
Archana Singh ⓘ https://orcid.org/0000-0003-1388-9489
Neeraj Dhar ⓘ https://orcid.org/0000-0002-5887-8137
Thomas John Pucadyil ⓘ https://orcid.org/0000-0002-2907-9889
Vinay Kumar Nandicoori ⓘ https://orcid.org/0000-0002-5682-4178

Reviewer #1 (Public review): https://doi.org/10.7554/eLife.104268.3.sa1
Reviewer #2 (Public review): https://doi.org/10.7554/eLife.104268.3.sa2
Reviewer #3 (Public review): https://doi.org/10.7554/eLife.104268.3.sa3
Author response https://doi.org/10.7554/eLife.104268.3.sa4

# Additional files

## Supplementary files

MDAR checklist

## Data availability

Source data used for generating data is provided in the manuscript.

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
