## [Editor Report · eLife Assessment]

Understanding bacterial growth mechanisms potentially uncover novel drug targets which are crucial for maintaining cellular viability, particularly for bacterial pathogens. In this **important** study, Kapoor et al, investigate the role of Wag31 in lipid and peptidoglycan biosynthesis in mycobacteria. A detailed analysis of Wag31 domain architecture revealed a role in membrane tethering. More specifically, the N-terminal and C-terminal domains appeared to have distinct functional roles. The data presented are **solid** and support the conclusion made. This study will be of broad interest to microbiologists and molecular biologists.

---

## [Referee Report · Reviewer #1 (Public review)]

This is a comprehensive study that sheds light on how Wag31 functions and localises in mycobacterial cells. A clear link to interactions with CL is shown using a combination of microscopy in combination with fusion fluorescent constructs, and lipid specific dyes. Furthermore, studies using mutant versions of Wag31 shed light on the functionalities of each domain in the protein. My concerns/suggestions for the manuscript are minor:

(1) Ln 130. A better clarification/discussion is required here. It is clear that both depletion and overexpression have an effect on levels of various lipids, but subsequent descriptions show that they affect different classes of lipids.

(2) The pulldown assays results are interesting, but the links are tentative.

(3) The authors may perhaps like to rephrase claims of effects lipid homeostasis, as my understanding is that lipid localisation rather than catabolism/breakdown is affected.

In response to the above reviews the authors have made the required changes in the revised manuscript.

---

## [Referee Report · Reviewer #2 (Public review)]

Summary:

Kapoor et. al. investigated the role of the mycobacterial protein Wag31 in lipid and peptidoglycan synthesis and sought to delineate the role of the N- and C- terminal domains of Wag31. They demonstrated that modulating Wag31 levels influences lipid homeostasis in *M. smegmatis* and cardiolipin (CL) localisation in cells. Wag31 was found to preferentially bind CL-containing liposomes, and deleting the N-terminus of the protein significantly decreased this interaction. Novel interactions between Wag31 and proteins involved in lipid metabolism and cell wall synthesis were identified, suggesting that Wag31 recruits proteins to the intracellular membrane domain by direct interaction.

Strengths:

(1) The importance of Wag31 in maintaining lipid homeostasis is supported by several lines of evidence.

(2) The interaction between Wag31 and cardiolipin, and the role of the N-terminus in this interaction was convincingly demonstrated.

Weakness:

(1) Interactome analysis with truncated versions of the proteins could not be performed in *M. smegmatis* due to protein instability.

---

## [Referee Report · Reviewer #3 (Public review)]

Summary:

This manuscript describes the characterization of mycobacterial cytoskeleton protein Wag31, examining its role in orchestrating protein-lipid and protein-protein interactions essential for mycobacterial survival. The most significant finding is that Wag31, which directs polar elongation and maintains the intracellular membrane domain, was revealed to have membrane tethering capabilities.

Strengths:

The authors provided a detailed analysis of Wag31 domain architecture, revealing distinct functional roles: the N-terminal domain facilitates lipid binding and membrane tethering, while the C-terminal domain mediates protein-protein interactions. Overall, this study offers a robust and new understanding of Wag31 function.

Weaknesses:

The authors did not address some of the comments. The following concerns should be addressed.

• As far as I can tell, authors did not address my prior comments on Line 270, which is Line 280 in the revised manuscript: the N-terminal region is important for lipid homeostasis, but the statement in Line 270, "the maintenance of lipid homeostasis by Wag31 is a consequence of its tethering activity" requires additional proof. Please indicate the page and line numbers in the revised manuscript so that I can identify the specific changes the authors made.

• Since this pull-down assay was conducted by mixing *E. coli* lysate expressing Wag31 and Msm lysate expression Wag31 interactors like MurG, it is possible that the interactions are not direct. Authors acknowledge that this is a valid point, and indicated that they "will describe this caveat in the revised manuscript". I have difficulty finding where this revision was made. Please indicate the page and line numbers.

---

## [Author Response]

The following is the authors’ response to the original reviews

**Public Reviews:**

**Reviewer #1 (Public review):**
This a comprehensive study that sheds light on how Wag31 functions and localises in mycobacterial cells. A clear link to interactions with CL is shown using a combination of microscopy in combination with fusion fluorescent constructs, and lipid specific dyes. Furthermore, studies using mutant versions of Wag31 shed light on the functionalities of each domain in the protein. My concerns/suggestions for the manuscript are minor:(1) Ln 130. A better clarification/discussion is required here. It is clear that both depletion and overexpression have an effect on levels of various lipids, but subsequent descriptions show that they affect different classes of lipids.

We thank the reviewer for the comment. We have added a better clarification on this in the discussion of revised manuscript. The lipid classes that get impacted by the depletion of Wag31 vs overexpression are different. Wag31 is an adaptor protein that interacts with proteins of the ACCase complex (Meniche et al., 2014; Xu et al., 2014) that synthesize fatty acid precursors and regulate their activity (Habibi Arejan et al., 2022).

The varied response on lipid homeostasis could be attributed to a change in the stoichiometry of these interactions of Wag31. While Wag31 depletion would prevent such interactions from occurring and might affect lipid synthesis that directly depends on Wag31-protein partner interactions, its overexpression would lead to promiscuous interactions and a change in the stoichiometry of native interactions that would ultimately modulate lipid synthesis pathways.

(2) The pulldown assays results are interesting, but links are tentative.

We thank the reviewer for the comment. The interactome of Wag31 was identified through the immunoprecipitation of FLAG-Wag31 complemented at an integrative locus in Wag31 mutant background to avoid overexpression artifacts. We used Msm::gfp expressing an integrative copy (at L5 locus) of FLAG-GFP as a control to subtract non-specific interactions. The experiment was performed in biological triplicates, and interactors that appeared in all replicates but not in the control were selected for further analysis. Although we identified more than 100 interactors of Wag31, we analyzed only the top 25 hits, with a PSM cut-off 18 and unique peptides5. Additionally, two of Wag31's established interactors, AccD5 and Rne, were among the top five hits, thus validating our data.

As mentioned in line 139 of the previous version of the manuscript, we agree that the interactions can either be direct or through a third partner. The fact that we obtained known interactors of Wag31 makes us believe these interactions are genuine. Moreover, for validation, we performed pulldown experiments by mixing *E. coli* lysates expressing His-Wag31 full-length or truncated protein with *M. smegmatis* lysates expressing FLAG-tagged interacting proteins. The wash conditions used were quite stringent for these pull-down assays—the wash buffer contained 1% Triton X100 that eliminates all non-specific and indirect interactions. However, we agree that we cannot conclusively state that the interactions are direct without purifying the proteins and performing the experiment. As mentioned above, this caveat was stated in the previous version of the manuscript.

(3) The authors may perhaps like to rephrase claims of effects lipid homeostasis, as my understanding is that lipid localisation rather than catabolism/breakdown is affected.

We thank the reviewer for the comment. In this manuscript, we are trying to convey that Wag31 is a spatiotemporal regulator of lipid metabolism. It is a peripheral protein that is hooked to the membrane via Cardiolipin and forms a scaffold at the poles, which helps localize several enzymes involved in lipid metabolism.

Homeostasis is the process by which an organism maintains a steady-state of balance and stability in response to changes. Depletion of Wag31 not only results in delocalisation of lipids in intracellular lipid inclusions but also leads to changes in the levels of various lipid classes. Advancement in the field of spatial biology underscores the importance of native localization of various biological molecules crucial for maintaining a steady-cell of the cell. Hence, we have used the word “homeostasis” to describe both the changes observed in lipid metabolism.

**Reviewer #2 (Public review):**
Summary:Kapoor et. al. investigated the role of the mycobacterial protein Wag31 in lipid and peptidoglycan synthesis and sought to delineate the role of the N- and C- terminal domains of Wag31. They demonstrated that modulating Wag31 levels influences lipid homeostasis in *M. smegmatis* and cardiolipin (CL) localisation in cells. Wag31 was found to preferentially bind CL-containing liposomes, and deleting the N-terminus of the protein significantly decreased this interaction. Novel interactions between Wag31 and proteins involved in lipid metabolism and cell wall synthesis were identified, suggesting that Wag31 recruits proteins to the intracellular membrane domain by direct interaction.Strengths:(1) The importance of Wag31 in maintaining lipid homeostasis is supported by several lines of evidence. (2) The interaction between Wag31 and cardiolipin, and the role of the N-terminus in this interaction was convincingly demonstrated.Weaknesses:(1) MS experiments provide some evidence for novel protein-protein interactions. However, the pulldown experiments lack a valid negative control.

We thank the reviewer for the comment. We have included two non-interactors of Wag31 i.e. MmpL4 and MmpS5 which were not identified in our interactome database as negative controls in the experiment. As shown in Figure S3, we performed His pull-down experiments with both of them independently twice, each time with a positive control (known interactor of Wag31 Msm2092). Fig. S3b revised shows *E. coli* lysate expressing His-Wag31 which was incubated with Msm lysates expressing either FLAG tagged-MmpL4 or -MmpS5 or Msm2092 (revised Fig. S3c). The mixed lysates were pulled down with Cobalt beads that bind to the His-tagged protein and analysed using Western blot analysis by probing with anti-FLAG antibody (revised Fig. S3d.). The data presented confirms that the interactions validated through the pull down assay were indeed specific.

(2) The role of the N-terminus in the protein-protein interaction has not been ruled out.

We thank the reviewer for the comment. Wag31_Msm_ is a 272 amino acids long protein. The Nterminal of Wag31, which houses the DivIVA-domain, comprises the first 60 amino acids. Previously, we attempted to express the N-terminal (60 aa long) and the C-terminal (212 aa long) truncated proteins in various mycobacterial shuttle vectors to perform MS/MS experiments. Despite numerous efforts, neither expressed with the N/C-terminal FLAG tag or no tag in episomal or integrative vectors due to instability of the protein. Eventually, we successfully expressed the C-terminal Wag31 with an N and Cterminal hexa-His tag. However, this expression was not sufficient or stable enough for us to perform Ni^2+^-affinity pull-down experiments for mass spectrometry. N-terminal of Wag31 could not be expressed in *M. smegmatis* even with N and C-terminal Hexa-His tags.

To rule out the role of the N-terminal in mediating protein-protein interactions, we cloned the N-terminal of Wag31 that comprises the DivIVA-domain in pET28b vector (Fig. 7a revised). Subsequently, the truncated protein, hereafter called Wag31_∆C_ flanked by 6X His tags at both the termini was expressed in *E. coli* and mixed with Msm lysates expressing interactors of Wag31 (Fig. 7b-c revised). Earlier experiments with Wag31_∆1-60</sub or Wag31∆N (in the revised manuscript) were performed with MurG, SepIVA, Msm2092 and AccA3 (Fig. 7e-g). Thus, we used the same set of interactors to test our hypothesis. Briefly, His- Wag31∆C was mixed with Msm lysates expressing either FLAG-MurG, -SepIVA, -Msm2092 or -AccA3 and pull down experiments were performed as described previously. FLAGMmpS5, a non-interactor of Wag31 was used as a negative control. As shown in Fig. 7d revised, His-Wag31 could bind to all the four interactors whereas His- Wag31∆C couldn’t, strengthening the conclusion that interactions of Wag31 with other proteins are mediated by its Cterminal. However, we can’t ignore the possibility of other interactors binding to the N-terminal of Wag31. Unfortunately, due to poor expression/instability of Wag31∆C in mycobacterial shuttle vectors, we are unable to perform a global interactome analysis of Wag31∆C_

**Reviewer #3 (Public review):**
Summary:This manuscript describes the characterization of mycobacterial cytoskeleton protein Wag31, examining its role in orchestrating protein-lipid and protein-protein interactions essential for mycobacterial survival. The most significant finding is that Wag31, which directs polar elongation and maintains the intracellular membrane domain, was revealed to have membrane tethering capabilities.Strengths:The authors provided a detailed analysis of Wag31 domain architecture, revealing distinct functional roles: the N-terminal domain facilitates lipid binding and membrane tethering, while the C-terminal domain mediates protein-protein interactions. Overall, this study offers a robust and new understanding of Wag31 function.Weaknesses:The following major concerns should be addressed.• Authors use 10-N-Nonyl-acridine orange (NAO) as a marker for cardiolipin localization. However, given that NAO is known to bind to various anionic phospholipids, how do the authors know that what they are seeing is specifically visualizing cardiolipin and not a different anionic phospholipid? For example, phosphatidylinositol is another abundant anionic phospholipid in mycobacterial plasma membrane.

We thank the reviewer for the comment. Despite its promiscuous binding to other anionic phospholipids, 10-N-Nonyl-acridine orange is widely used to stain Cardiolipin and determine its localisation in bacterial cells and mitochondria of eukaryotes (Garcia Fernandez et al., 2004; Mileykovskaya & Dowhan, 2000; Renner & Weibel, 2011). This is because it has a stronger affinity for Cardiolipin than other anionic phospholipids with the affinity constant being 2 × 10^6^ M−^1^ for Cardiolipin association and 7 × 10^4^ M−^1^ for that of phosphatidylserine and phosphatidylinositol association (Petit et al., 1992). Additionally, there is not yet another stain available for detecting Cardiolipin. Our proteinlipid binding assays suggest that Wag31 preferentially binds to Cardiolipin over other anionic phospholipids (Fig. 4b), hence it is likely that the majority of redistribution of NAO fluorescence that we observe might be contributed by Cardiolipin mislocalization due to altered Wag31 levels, with smaller degree of NAO redistribution intensity coming indirectly from other anionic phospholipids displaced from the membrane due to the loss of membrane integrity and cell shape changes due to Wag31.

• Authors' data show that the N-terminal region of Wag31 is important for membrane tethering. The authors' data also show that the N-terminal region is important for sustaining mycobacterial morphology. However, the authors' statement in Line 256 "These results highlight the importance of tethering for sustaining mycobacterial morphology and survival" requires additional proof. It remains possible that the N-terminal region has another unknown activity, and this yet-unknown activity rather than the membrane tethering activity drives the morphological maintenance. Similarly, the N-terminal region is important for lipid homeostasis, but the statement in Line 270, "the maintenance of lipid homeostasis by Wag31 is a consequence of its tethering activity" requires additional proof. The authors should tone down these overstatements or provide additional data to support their claims.

We agree with the reviewer that there exists a possibility for another function of the N-terminal that may contribute to sustaining mycobacterial physiology and survival. We would revise our statements in the paper to reflect the data. Results shown suggest that the tethering activity of the Nterminal region may contribute to mycobacterial morphology and survival. However, additional functions of this region can’t be ruled out. Similarly, the maintenance of lipid homeostasis by Wag31 may be associated with its tethering activity, although other mechanisms could also contribute to this process.

• Authors suggest that Wag31 acts as a scaffold for the IMD (Fig. 8). However, Meniche et. al. has shown that MurG as well as GlfT2, two well-characterized IMD proteins, do not colocalize with Wag31 (DivIVA) (https://doi.org/10.1073/pnas.1402158111). IMD proteins are always slightly subpolar while Wag31 is located to the tip of the cell. Therefore, the authors' biochemical data cannot be easily reconciled with microscopic observations in the literature. This raises a question regarding the validity of protein-protein interaction shown in Figure 7. Since this pull-down assay was conducted by mixing *E. coli* lysate expressing Wag31 and Msm lysate expression Wag31 interactors like MurG, it is possible that the interactions are not direct. Authors should interpret their data more cautiously. If authors cannot provide additional data and sufficient justifications, they should avoid proposing a confusing model like Figure 8 that contradicts published observations.

In the literature, MurG and GlfT2 have been shown to have polar localisation (Freeman et al., 2023; Hayashi et al., 2016; Kado et al., 2023) and two groups have shown slightly sub-polar localisation of MurG (García-Heredia et al., 2021; Meniche et al., 2014). Additionally, (Freeman et al., 2023) showed SepIVA to be a spatio-temporal regulator of MurG. MS/MS analysis of Wag31 immunoprecipitation data yielded both MurG and SepIVA to be interactors of Wag31 (Fig. 3). Given Wag31 also displays polar localisation, it is likely that it associates with the polar MurG. However, since a sub-polar localisation of MurG has also been reported, it is possible that they do not interact directly and another protein mediates their interaction. Based on the above, we will modify the model proposed in Fig. 8.

We agree that for validation of interaction, we performed pulldown experiments by mixing *E. coli* lysates expressing His-Wag31 full-length or truncated protein with M. smegmatis lysates expressing FLAG-tagged interacting proteins. The wash conditions used were quite stringent for these pull-down assays—the wash buffer contained 1% Triton X100 that eliminates all non-specific and indirect interactions. However, we agree that we cannot conclusively state that the interactions are direct without purifying the proteins and performing the experiment. We will describe this caveat in the revised manuscript and propose a model that reflects the results we obtained.

References:

Freeman, A. H., Tembiwa, K., Brenner, J. R., Chase, M. R., Fortune, S. M., Morita, Y. S., & Boutte, C. C. (2023). Arginine methylation sites on SepIVA help balance elongation and septation in *Mycobacterium smegmatis*. Mol Microbiol, 119(2), 208-223. https://doi.org/10.1111/mmi.15006

Garcia Fernandez, M. I., Ceccarelli, D., & Muscatello, U. (2004). Use of the fluorescent dye 10-N-nonyl acridine orange in quantitative and location assays of cardiolipin: a study on different experimental models. Anal Biochem, 328(2), 174-180. https://doi.org/10.1016/j.ab.2004.01.020

García-Heredia, A., Kado, T., Sein, C. E., Puffal, J., Osman, S. H., Judd, J., Gray, T. A., Morita, Y. S., & Siegrist, M. S. (2021). Membrane-partitioned cell wall synthesis in mycobacteria. eLife, 10. https://doi.org/10.7554/eLife.60263

Habibi Arejan, N., Ensinck, D., Diacovich, L., Patel, P. B., Quintanilla, S. Y., Emami Saleh, A., Gramajo, H., & Boutte, C. C. (2022). Polar protein Wag31 both activates and inhibits cell wall metabolism at the poles and septum. Front Microbiol, 13, 1085918. https://doi.org/10.3389/fmicb.2022.1085918

Hayashi, J. M., Luo, C. Y., Mayfield, J. A., Hsu, T., Fukuda, T., Walfield, A. L., Giffen, S. R., Leszyk, J. D., Baer, C. E., Bennion, O. T., Madduri, A., Shaffer, S. A., Aldridge, B. B., Sassetti, C. M., Sandler, S. J., Kinoshita, T., Moody, D. B., & Morita, Y. S. (2016). Spatially distinct and metabolically active membrane domain in mycobacteria. Proc Natl Acad Sci U S A, 113(19), 5400-5405. https://doi.org/10.1073/pnas.1525165113

Kado, T., Akbary, Z., Motooka, D., Sparks, I. L., Melzer, E. S., Nakamura, S., Rojas, E. R., Morita, Y. S., & Siegrist, M. S. (2023). A cell wall synthase accelerates plasma membrane partitioning in mycobacteria. eLife, 12, e81924. https://doi.org/10.7554/eLife.81924

Meniche, X., Otten, R., Siegrist, M. S., Baer, C. E., Murphy, K. C., Bertozzi, C. R., & Sassetti, C. M. (2014). Subpolar addition of new cell wall is directed by DivIVA in mycobacteria. Proc Natl Acad Sci U S A, 111(31), E32433251. https://doi.org/10.1073/pnas.1402158111

Mileykovskaya, E., & Dowhan, W. (2000). Visualization of phospholipid domains in *Escherichia coli* by using the cardiolipin-specific fluorescent dye 10-N-nonyl acridine orange. J Bacteriol, 182(4), 1172-1175. https://doi.org/10.1128/JB.182.4.1172-1175.2000

Petit, J. M., Maftah, A., Ratinaud, M. H., & Julien, R. (1992). 10N-nonyl acridine orange interacts with cardiolipin and allows the quantification of this phospholipid in isolated mitochondria. Eur J Biochem, 209(1), 267273. https://doi.org/10.1111/j.1432-1033.1992.tb17285.x

Renner, L. D., & Weibel, D. B. (2011). Cardiolipin microdomains localize to negatively curved regions of *Escherichia coli* membranes. Proc Natl Acad Sci U S A, 108(15), 6264-6269. https://doi.org/10.1073/pnas.1015757108

Schägger, H. (2006). Tricine-SDS-PAGE. Nat Protoc, 1(1), 16-22. https://doi.org/10.1038/nprot.2006.4

Xu, W. X., Zhang, L., Mai, J. T., Peng, R. C., Yang, E. Z., Peng, C., & Wang, H. H. (2014). The Wag31 protein interacts with AccA3 and coordinates cell wall lipid permeability and lipophilic drug resistance in *Mycobacterium smegmatis*. Biochem Biophys Res Commun, 448(3), 255-260. https://doi.org/10.1016/j.bbrc.2014.04.116

**Recommendations for the authors:**

**Reviewer #1 (Recommendations for the authors):**
(1) Ln 130. A better clarification/discussion is required here. It is clear that both depletion and overexpression have an effect in levels of various lipids, but subsequent descriptions show that they affect different classes of lipids.

We thank the reviewer for the comment. We have included a clarification for this in the discussion section.

(2) The pulldown assays results are interesting, but the links are tentative.

We thank the reviewer for the comment. The interactome of Wag31 was identified through the immunoprecipitation of Flag-tagged Wag31 complemented at an integrative locus in Wag31 mutant background to avoid overexpression artifacts. We used Msm::gfp expressing an integrative copy (at L5 locus) of FLAG-GFP as a control to subtract non-specific interactions. The experiment was performed in biological triplicates, and interactors that appeared in all replicates were selected for further analysis. Although we identified more than 100 interactors of Wag31, we analyzed only the top 25 hits, with a PSM cut-off 18 and unique peptides5. Additionally, two of Wag31's established interactors, AccD5 and Rne, were among the top five hits, thus validating our data.

Though we agree that the interactions can either be direct or through a third partner, the fact that we obtained known interactors of Wag31 makes us believe these interactions are genuine. Moreover, for validation, we performed pulldown experiments by mixing *E. coli* lysates expressing HisWag31 full-length or truncated protein with M. smegmatis lysates expressing FLAG-tagged interacting proteins. The wash conditions used were quite stringent for these pull-down assays—the wash buffer contained 1% Triton X100 that eliminates all non-specific and indirect interactions. However, we agree that we cannot conclusively state that the interactions are direct without purifying the proteins and performing the experiment. We will describe this caveat in the revised manuscript.

(3) The authors may perhaps like to rephrase claims of effects lipid homeostasis, as my understanding is that lipid localisation rather than catabolism/breakdown is affected.

We thank the reviewer for the comment. In this manuscript, we are trying to convey that Wag31 is a spatiotemporal regulator of lipid metabolism. It is a peripheral protein that is hooked to the membrane via Cardiolipin and forms a scaffold at the poles, which helps localize several enzymes involved in lipid metabolism.

Homeostasis is the process by which an organism maintains a steady-state of balance and stability in response to changes. Depletion of Wag31 not only results in delocalisation of lipids in intracellular lipid inclusions but also leads to changes in the levels of various lipid classes. Advancement in the field of spatial biology underscores the importance of native localization of various biological molecules crucial for maintaining a steady-cell of the cell. Hence, we have used the word “homeostasis” to describe both the changes observed in lipid metabolism.

**Reviewer #2 (Recommendations for the authors):**
I recommend the following experiments to strengthen the data presented:(1) Include a non-interacting FLAG-tagged protein as a negative control in the pull-down experiment to strengthen this data.

We thank the reviewer for the comment. As suggested, we have included non-interacting FLAGtagged proteins as negative controls in the pulldown experiment. We chose MmpL4 and MmpS5 which were not found in the Wag31 interactome data. We performed pull-down experiments with both of them and included an interactor of Wag31 i.e. Msm2092 as a positive control. Fig. S3b revised shows *E. coli* lysate expressing His-Wag31 which was incubated with Msm lysates expressing either FLAG taggedMmpL4 or -MmpS5 or -Msm2092 (Fig. S3c revised). The mixed lysates were pulled down with Cobalt beads that bind to the His-tagged protein and analysed using Western blot analysis by probing with anti-FLAG antibody. The pull down experiments were performed independently twice, every time with Msm2092 as the positive control (Fig. S3d. revised).

(2) Perform the pull-down experiments using only the Wag31 N-terminus to rule out any role that it may have in the protein-protein interactions.

We thank the reviewer for the comment. To rule out the possibility of N-terminal of Wag31 in mediating protein-protein interactions, we cloned the N-terminal of Wag31 that comprises the DivIVAdomain in pET28b vector (Fig. 7a revised). Subsequently, the truncated protein, hereafter called Wag31_∆C_ flanked by 6X His tags at both the termini was expressed in *E. coli* and subsequently mixed with Msm lysates expressing interactors of Wag31 (Fig. 7b-c revised). Earlier experiments with Wag31_∆1-60_ or Wag31_∆N_ were performed with MurG, SepIVA, Msm2092 and AccA3 (Fig. 7 previous) so we used the same set of interactors to test our hypothesis. Briefly, His-Wag31_∆C_was mixed with Msm lysates expressing either FLAG-MurG, -SepIVA, -Msm2092 or -AccA3 and pull down experiments were performed as described previously. FLAG-MmpS5, a non-interactor of Wag31 was used as a negative control. As shown in Fig. 7d revised, His-Wag31 could bind to all the four interactors whereas His-Wag31_∆C_ couldn’t, strengthening the conclusion that interactions of Wag31 with other proteins are mediated by its C-terminal. However, we can’t ignore the possibility of other proteins binding to the Nterminal of Wag31. Unfortunately, due to poor expression/instability of Wag31_∆C_ in mycobacterial shuttle vectors, we couldn’t perform a global interactome analysis of Wag31_∆C_.

Minor comments:- Please check the legend of Fig. 1g, it appears to be labelled incorrectly.

We have checked it. It is correct. From Fig. 1g we are trying to reflect on the percentages of cells of the three strains i.e. *Msm*+ATc, *Δwag31-*ATc, and *Δwag31*+ATc displaying rod, round or bulged morphology.

- For MS/MS analysis, a GFP control is mentioned but it is not indicated how this was incorporated in the data analysis. This information should be added.

We have incorporated that in the revised methodology.

- The information presented in Fig. 3a, e and f could be combined in one table.

We appreciate the idea of the reviewer but we prefer a pictorial representation of the data. It allows readers to consume the information in parts, make quicker comparisons and understand trends easily.

- Fig. 4c Wag31K20A appears smaller in size than the wild-type protein - why is this the case? Is this not a single amino acid substitution?

Though K20A is a single amino acid substitution, it alters the mobility of Wag31 on SDS-PAGE gel. The sequence analysis of the plasmid expressing Wag31_K20A_ doesn’t show additional mutations other than the desired K20A. The change in mobility could be due to a change in the conformation of Wag31_K20A_ or its ability to bind to SDS or both that modify its mobility under the influence of electric field.

- Please clarify what is contained in the first panel of fig 4e. compared to what is in the second panel.

The first panel represents CL-Dil-Liposomes before incubation with Wag31-GFP and the second panel shows CL-Dil-Liposomes after incubation with Wag31-GFP. The third panel shows the mixture as observed in the green channel to investigate the localisation of Wag31-GFP in the liposome-protein mix. Fourth panel shows the merged of second and third.

- The data in Fig 6d suggests higher levels of CL in the ∆wag31 compared to wild-type - how do the authors reconcile this with the MS data in Fig. 2g showing lower CL levels?

Fig. 6d represents the distribution of CL localisation in the tested strains of mycobacteria whereas Fig. 2g shows the absolute levels of CL in various strains. We attribute greater confidence on the lipidomics data which suggests down regulation of CL species. The NAO staining and microscopy is merely for studying localization of the CL along the cell, and cannot be used to reliably quantify or equate it to CL levels. The staining using a probe such as NAO is dependent on factors such as hydrophobicity and permeability of the cell wall, which we expect to be severely altered in a Wag31 mutant. Therefore, the increased staining of NAO seen in Wag31 mutant could just be reflective of the increased uptake of the dye rather than absolute levels of CL. The specificity of staining and localization however can be expected to be unaltered.

**Reviewer #3 (Recommendations for the authors):**
Following are suggestions for improving the writing and presentation.• Figure 1, the meaning of the yellow arrows present in f and h should be mentioned in the figure legend.

We have incorporated that in the revised legend. In Fig.1f, the yellow arrowhead represents the bulged pole morphology whereas in Fig. 1h, it indicates intracellular lipid inclusions.

• Figure 7 legend refers to panels g, h, and i. However, Figure 7 only has panels a-c. The legend lacks a description of panel c.

We have corrected the typos and the legend.

• Figure S1, F2-R2 and F3-R3 expected sizes should be stated in the legend of the figure.

We have updated the legends.

• Figure S5, is this the same figure as 5e? If so, there is no need for this figure.

We have removed Fig. S5.

• Methods need to be written more carefully with enough details. I listed some of the concerns below.

Detailed methodology was previously provided in the supplementary material and now we have moved it to the materials and methods in the revised manuscript.

• Line 392, provide more details on western blotting. What is the secondary antibody? What image documentation system was used?

We have updated the methodology.

• Line 400, while the methods may be the same as the reference 64, authors should still provide key details such as the way samples were fixed and processed for SEM and TEM.

We have provided a detailed description of the same in methodology in the revised version.

• Line 437, how do authors calculate the concentration of liposome to be 10 µM? Do they possibly mean the concentration of phospholipids used to make the liposomes?

Yes, this is the concentration of total lipids used to make liposomes. 1 μM of Wag31 or its mutants were mixed with 100 nm extruded liposomes containing 10 μm total lipid in separate Eppendorf tubes.

• Supplemental Line 9, "turns of" should read "turns off".

We have edited this.

• Supplemental Line 13, define LHS and RHS.

LHS or left hand sequence and RHS or right hand sequence refers to the upstream and downstream flanking regions of the gene of interest.

• Supplemental Line 20, indicate the manufacturer of the microscope and type of the objective lens.

We have added these details now.

• Supplemental Line 31, define MeOH, or use a chemical formula like chloroform.

MeOH is methanol. We have provided a chemical formula in the revised version.

• Supplemental Line 53, indicate the concentration of trypsin.

We have included that in the revised version.

• Supplemental Line 72, g is not a unit. "30,000 g" should be "30,000x g".

We have revised this in the manuscript.

• Supplemental Line 114, provide more details on western blotting. What is the manufacturer of antiFLAG antibody? What is the secondary antibody? How was the antibody binding visualized? What image documentation system was used?

We have provided these details in the revised version.